# Knowledge Externalization: Reversible Unlearning and Modular Retrieval in Multimodal Large Language Models

**Jiaqi Li**[1,3][*] **Zihan You**[2][*] **Ruoyan Shen**[3,4]**, ShenYu Zhang**[3,4]**, Songlin Zhai**[3,4]**, Yongrui Chen**[3,4]**,
Chuanyi Zhang**[8]**, Jiahui Geng**[6,7]**, Fakhri Karray**[6]**, Sheng Bi**[5]**, Guilin Qi**[3,4][†]

[1]School of Cyber Science and Engineering, Southeast University, Nanjing, China
[2]School of Instrument Science and Engineering, Southeast University, Nanjing, China
[3]Key Laboratory of New Generation Artificial Intelligence Technology and Its
Interdisciplinary Applications (Southeast University), Ministry of Education, China
[4]School of Computer Science and Engineering, Southeast University, Nanjing, China
[5]Law and Innovation Lab, School of Law, Southeast University
[6]Mohamed bin Zayed University of Artificial Intelligence
[7]Linköping University
[8]College of Artificial Intelligence and Automation, Hohai University, Nanjing, China
{jq1i,213223318,220242389,shenyu.zhang,songlin_zhai,yrchen,shengbi,gqi}@seu.edu.cn
20231104@hhu.edu.cn, jiahui.geng@liu.se, {jiahui.geng,Fakhri.Karray}@mbzuai.ac.ae,

## Abstract

Multimodal Large Language Models (MLLMs) achieve remarkable cross-modal understanding by training on vast web-scale datasets, but inadvertently internalize sensitive personal and proprietary information. Existing machine unlearning methods address this by irreversibly altering model parameters to permanently erase knowledge. This destructive paradigm conflicts with modern privacy regulations that mandate auditable, reversible, and user-controllable data management. To address these challenges, we propose Knowledge Externalization, a novel framework for reversible and modular knowledge management in MLLMs. We first propose Dual-Stream Memory Tuning, a method that transfers targeted knowledge from a model's internal parameters into external memory tokens. To mitigate gradient interference when externalizing multiple concepts, we further introduce Soft Orthogonal Weighting, a technique that preserves the independence of each token. Our resulting framework demonstrates three key capabilities: (i) It achieves effective forgetting of target concepts within the base model, while enabling high-fidelity knowledge restoration using the corresponding memory token. (ii) It supports continuous Knowledge Editing, allowing the knowledge stored within an external token to be dynamically updated post-externalization. (iii) It displays a remarkable emergent ability for compositionality, where multiple memory tokens (including edited ones) can be freely combined to simultaneously recover knowledge corresponding to each concept. Our source code could be available at https://github.com/ZihanYou/Knowledge_Externalization.

## 1 Introduction

The emergence of Multimodal Large Language Models (MLLMs) marks a significant breakthrough in artificial intelligence (Chen et al., 2024a; Li et al., 2023; Koh et al., 2023; Dai et al., 2023; Fang et al., 2025a;c), enabling advanced cross-modal understanding and generation across domains like medical imaging interpretation and visual storytelling (Zheng et al., 2023; Huang et al., 2024; Zhang et al., 2024a; Huang et al., 2023). Despite their impressive capabilities, MLLMs pose critical privacy risks by inadvertently internalizing sensitive personal data and proprietary information during pretraining

---

[*]  J. Li and Z. You contributed equally to this work and should be considered co-first authors.
[†]  Corresponding author.

(Mantelero, 2013; Scherer & Kiparski, 2018; Leite et al., 2022). To mitigate these risks, machine unlearning techniques (Eldan & Russinovich, 2023; Si et al., 2023; Wang et al., 2023; Thaker et al., 2024; Liu et al., 2024) have gained prominence, selectively forgetting targeted knowledge from models while maintaining utility without requiring costly retraining.

However, current unlearning methods inherently rely on irreversible parameter modifications (Yao et al., 2023; 2024; Gandikota et al., 2024; Lu et al., 2024), permanently erasing target knowledge from the model. This paradigm contradicts emerging privacy regulations like ISO/IEC 27701[1], which emphasize reversible, auditable, and user-controllable management of personally identifiable information (PII). Specifically, these frameworks advocate for Privacy Information Management Systems (Tzolov, 2019; Anwar & Gill, 2021) that empower users with granular control, such as the right to restrict processing (GDPR Art. 18) (Scherer & Kiparski, 2018) rather than only permanent deletion, and demand auditable logs of all data management activities. This creates a need for systems that can not only forget, but also manage and trace knowledge in a reversible manner. To address these challenges, we introduce *Knowledge Externalization*, a novel framework for reversible and modular knowledge management in MLLMs. Inspired by the concept of the 'Pensieve' from the Harry Potter universe, which enables temporary removal and later retrieval of memories, our approach transfers target knowledge from internal model parameters to external memory tokens. This decoupling allows the base model to forget sensitive information, while preserving the ability to restore it via memory retrieval when needed. In addition, by externalizing knowledge into dedicated memory tokens, the forgetting process remains localized, helping to reduce unintended effects on non-target model behavior.

To implement the paradigm, we propose a baseline method called *Dual-Stream Memory Tuning (DSM)*. DSM uses a gradient-ascent objective to erase targeted knowledge from the base model, while concurrently applying gradient-descent to encode the removed knowledge into external memory tokens. In practical scenarios, it is often necessary to remove and manage multiple knowledge concepts, such as different personal entities or domain-specific facts. To support this, DSM assigns a dedicated memory token to each concept and ensures that only the corresponding token is updated during externalization.

However, this multi-concept setup introduces a key challenge: when different memory tokens are trained independently, their gradient updates may influence overlapping model parameters, leading to interference that degrades the fidelity of each token. To mitigate this, we propose *Soft Orthogonal Weighting (SOW)*, which replaces hard gradient masking with an exponential attenuation scheme. Specifically, when encoding a new concept, we measure its gradient similarity to historical concepts and apply $w(s) = e^{-\lambda(s+1)}$ to attenuate interference while preserving optimization flow.

For evaluating Knowledge Externalization, we extend MMUBench (Li et al., 2024b) into MXEBench, a new benchmark for Multimodal Knowledge Externalization. Results demonstrate that our method achieves effective forgetting of target concepts, minimal degradation on non-target knowledge, and precise recovery via memory tokens. Beyond reversible unlearning, we also show that our externalized memory design supports Dynamic Knowledge Editing (detailed in Sec.4.3) and Emergent Compositional Knowledge Recovery (detailed in Sec.4.4), which are opening new possibilities for controllable and modular knowledge systems.

In summary, our contributions are: **(i)** We propose *Knowledge Externalization*, the first reversible framework for managing unlearning knowledge in MLLMs via external memory tokens. **(ii)** We introduce *Dual-Stream Memory Tuning* to decouple knowledge unlearning from knowledge retention using dual-gradient optimization. **(iii)** We design *Soft Orthogonal Weighting* with exponential attenuation, enabling scalable multi-concept externalization with provable interference bounds. **(iv)** We empirically demonstrate high-fidelity knowledge removal, restoration, editing, and composition across diverse concepts.

## 2 RELATED WORK

**Unlearning in MLLMs.** Recent Machine Unlearning research has explored various strategies for LLMs and MLLMs (Jang et al., 2023; Kumar et al., 2023; Pawelczyk et al., 2024; Ishibashi &

---

[1]ISO/IEC 27701 extends the ISO/IEC 27001 standard to include privacy management, providing a framework for organizations to manage personally identifiable information (PII).

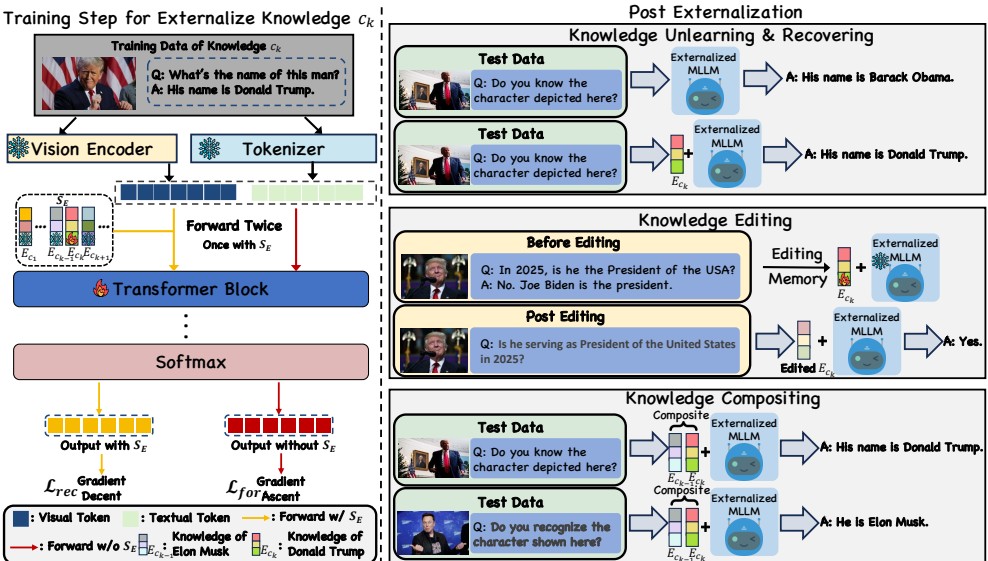

Figure 1: **Overview of the Knowledge Externalization Framework. Left:** The training process of **Dual-Stream Memory Tuning** utilizes parallel forward passes to simultaneously erase knowledge from the base model (via gradient ascent on $\mathcal{L}_{\text{for}}$) and encode it into an external memory token $E_{C_k}$ (via gradient descent on $\mathcal{L}_{\text{rec}}$). **Right:** This modular design enables key post-externalization capabilities. **(Top)** The base model forgets the concept, while knowledge is recovered with the token. **(Middle)** Knowledge can be dynamically updated by editing the token alone. **(Bottom)** Multiple tokens can be composed at inference to recover all corresponding knowledge, which is a unique emergent capability of our framework.

Shimodaira, 2023; Maini et al., 2024; Thaker et al., 2024; Liu et al., 2024), including Gradient Ascent to eliminate undesirable outputs (Yao et al., 2023), knowledge alignment approaches (Wang et al., 2023), lightweight unlearning layers (Chen & Yang, 2023), and methods combining GA with KL-divergence (Yao et al., 2024). SIU (Li et al., 2024b) focuses on erasing visual concepts while preserving generative capabilities. However, these approaches implement irreversible parameter modifications that permanently erase information. Our **Knowledge Externalization** framework introduces reversible unlearning by transferring target knowledge to external memory tokens, enabling precise recovery, compositional querying, and dynamic editing.

**Parameter-Efficient Fine-Tuning.** PEFT methods reduce fine-tuning costs through *adapter modules* that insert small networks into transformer layers (Gao et al., 2024; Luo et al., 2024; Ran et al., 2023; Guo et al., 2024), *Low-Rank Adaptation (LoRA)* that approximates updates via low-rank matrices (Hu et al., 2022) and is widely adopted across vision (Hayou et al., 2024; Agiza et al., 2024), speech (Prasad et al., 2024; Baby et al., 2024), and multimodal domains (Guo et al., 2025; Wu et al., 2024), and *prompt-based tuning* including prefix tuning (Li & Liang, 2021), prompt tuning (Jia et al., 2022), and multimodal extensions (Zhou et al., 2024; Shen et al., 2024). These methods target **task adaptation** by efficiently *adding* knowledge, while our approach **encodes multiple concepts into separate, controllable modules**. Unlike adapters or LoRA that modify internal paths, our memory tokens operate as **external, composable, and reversible** modules.

# 3 TASK DEFINITION

Let $\mathcal{M}_\theta$ denotes a pretrained Multimodal Large Language Model (MLLM) with parameters $\theta \in \mathbb{R}^d$, trained on a multimodal dataset $\mathcal{D} = \{(\mathcal{I}_i, \mathcal{T}_i)\}_{i=1}^N$ where $\mathcal{I}_i \in \mathbb{R}^{H \times W \times 3}$ are images and $\mathcal{T}_i = (w_1^i, \dots, w_{t_i}^i)$ are text sequences. During pretraining, $\mathcal{M}_\theta$ may internalize sensitive concepts $\mathcal{C} = \{c_1, \dots, c_k\}$, each associated with data $\mathcal{D}_{c_k} \subseteq \mathcal{D}$.

We define the goal of Knowledge Externalization in MLLMs as follows:

> **Knowledge Externalization**
>
> **Knowledge Externalization** enables the temporary removal of sensitive knowledge from MLLMs without degrading its utility, while retaining the capability to recover that knowledge through the external memory.

The optimization objective can be formulated as follows:

$$
\min_{\theta', \mathcal{S}_E} \quad \underbrace{\sum_{c_k \in \mathcal{C}} \mathbb{E}_{(\mathcal{I}, \mathcal{T}) \in \mathcal{D}_{c_k}} \left[ \sum_{t=1}^{T} \log P_{\mathcal{M}_{\theta'}}(w_t \mid \mathcal{I}, w_{<t}) \right]}_{\text{Forgetting loss } \mathcal{L}_{\text{for}}} + \lambda \underbrace{\sum_{(\mathcal{I}, \mathcal{T}) \in \mathcal{D} \setminus \cup_k \mathcal{D}_{c_k}} \left[ -\sum_{t=1}^{T} \log P_{\mathcal{M}_{\theta'}}(w_t \mid \mathcal{I}, w_{<t}) \right]}_{\text{Utility preservation loss } \mathcal{L}_{\text{pre}}}
$$

$$
+ \mu \underbrace{\sum_{c_k \in \mathcal{C}} \mathbb{E}_{(\mathcal{I}, \mathcal{T}) \in \mathcal{D}_{c_k}} \left[ -\sum_{t=1}^{T} \log P_{\mathcal{M}_{\theta'}}(w_t \mid [E_{c_k}; \mathcal{I}, w_{<t}]) \right]}_{\text{Recoverability loss } \mathcal{L}_{\text{rec}}},
$$

$$(1)$$

where $\mathcal{S}_E = \{E_{c_k}\}_{c_k \in \mathcal{C}}$ is the set of externalized memory tokens, and $[E_{c_k}; \mathcal{I}, w_{<t}]$ denotes the concatenation of $E_{c_k}$ as a prefix to the multimodal input sequence.

## 4 METHODOLOGY

To achieve the objectives of Knowledge Externalization defined in Eq.1, we propose **Dual-Stream Memory Tuning (DSM)**, which is a framework that decouples the forgetting of sensitive concepts from their recoverable preservation (presented in Fig.1). DSM operates by simultaneously optimizing the base model parameters $\theta'$ and a set of external memory tokens $\mathcal{S}_E = \{E_{c_k}\}_{c_k \in \mathcal{C}}$, ensuring that knowledge is not erased but *transferred* from internal parameters to modular, auditable tokens. While DSM naturally supports single-concept externalization, managing multiple concepts introduces gradient interference, where updates for one concept degrade the fidelity of others. To enable scalable and modular knowledge management, we further introduce **Soft Orthogonal Weighting**, a regularization mechanism that encourages near-orthogonal update directions in the parameter space.

### 4.1 DUAL-STREAM MEMORY TUNING

The core mechanism of DSM is a dual-stream optimization process that implements a zero-sum game between forgetting and recovery. For each concept $c_k$, the base model $\mathcal{M}_{\theta'}$ undergoes **gradient ascent** on the forgetting loss $\mathcal{L}_{\text{for}}$:

$$
\theta' \leftarrow \theta' + \eta \cdot \nabla_{\theta'} \mathcal{L}_{\text{for}} = \theta' + \eta \cdot \nabla_{\theta'} \mathbb{E}_{(\mathcal{I}, \mathcal{T}) \in \mathcal{D}_{c_k}} \left[ \sum_{t=1}^{T} \log P_{\mathcal{M}_{\theta'}}(w_t \mid \mathcal{I}, w_{<t}) \right], \tag{2}
$$

which repels $\theta'$ from the knowledge manifold of $c_k$, effectively erasing its influence on model predictions.

Concurrently, to preserve this knowledge for authorized recovery, we perform **gradient descent** on the recoverability loss $\mathcal{L}_{\text{rec}}$ with respect to both the memory token $E_{c_k}$ and the model parameters $\theta'$:

$$
E_{c_k} \leftarrow E_{c_k} - \gamma \cdot \nabla_{E_{c_k}} \mathcal{L}_{\text{rec}}, \tag{3}
$$

$$
\theta' \leftarrow \theta' - \gamma \cdot \nabla_{\theta'} \mathcal{L}_{\text{rec}}, \tag{4}
$$

where

$$
\mathcal{L}_{\text{rec}} = \mathbb{E}_{(\mathcal{I}, \mathcal{T}) \in \mathcal{D}_{c_k}} \left[ \sum_{t=1}^{T} \log P_{\mathcal{M}_{\theta'}}(w_t \mid [E_{c_k}; \mathcal{I}], w_{<t}) \right]. \tag{5}
$$

### 4.2 SOFT ORTHOGONAL WEIGHTING FOR MULTI-CONCEPT EXTERNALIZATION

For multi-concept externalization, the gradient updates for different concepts may interfere when they share parameter subspaces, leading to degraded token fidelity. To mitigate this, we introduce **Soft Orthogonal Weighting (SOW)**, which dynamically attenuates the memory token update based on its similarity to historical recovery directions. As shown in Fig.2, SOW maintains a gradient history dictionary $\mathcal{H} = \{c_j : \mathbf{g}_j\}_{j=1}^{k-1}$, where

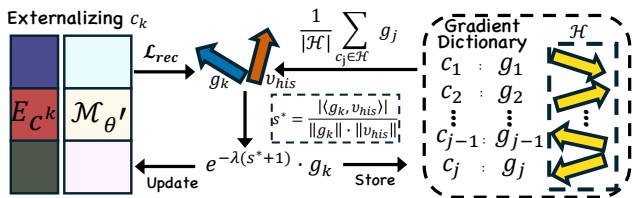

Figure 2: **Illustration of Soft Orthogonal Weighting (SOW).** When externalizing a new concept $c_k$, its recovery gradient $\mathbf{g}_k$ is compared against a historical composite gradient $\mathbf{v}_{his}$ synthesized from the Gradient Dictionary $\mathcal{H}$. The resulting cosine similarity $s^*$ is used to compute an exponential weight $w(s^*)$, which attenuates the update applied to the memory token $E_{c_k}$ and the model $\mathcal{M}_{\theta'}$.

$\mathbf{g}_j = \nabla_{\theta'}\mathcal{L}_{\text{rec}}(c_j)$ is the recovery gradient of concept $c_j$. For a new concept $c_k$, we first compute its raw gradient $\mathbf{g}_k$, then construct a *historical composite gradient*:

$$\mathbf{v}_{\text{his}} = \sum_{c_j \in \mathcal{H}} \alpha_j \mathbf{g}_j, \quad \alpha_j = \frac{\|\mathbf{g}_j\|}{\sum_{c_i \in \mathcal{H}} \|\mathbf{g}_i\|}, \tag{6}$$

which represents the dominant direction of prior knowledge unlearning. We then compute the cosine similarity between $\mathbf{g}_k$ and $\mathbf{v}_{\text{hist}}$:

$$s^* = \frac{|\langle \mathbf{g}_k, \mathbf{v}_{\text{his}} \rangle|}{\|\mathbf{g}_k\| \cdot \|\mathbf{v}_{\text{his}}\|}. \tag{7}$$

This similarity reflects how much the new concept overlaps with prior recovery directions. We define an attenuation weight:

$$w(s^*) = e^{-\lambda(s^*+1)}, \quad \lambda > 0, \tag{8}$$

which suppresses updates in redundant directions. The final updates are:

$$E_{c_k} \leftarrow E_{c_k} - \gamma \cdot w(s^*) \cdot \nabla_{E_{c_k}} \mathcal{L}_{\text{rec}}, \tag{9}$$

$$\theta' \leftarrow \theta' - \gamma \cdot w(s^*) \cdot \nabla_{\theta'} \mathcal{L}_{\text{rec}}. \tag{10}$$

The theoretical analysis of SOW is presented in Sec.A.4.

## 4.3 DYNAMIC KNOWLEDGE EDITING

Knowledge editing (Li & Chu, 2025; Xu et al., 2025; Zhang et al., 2024b) is critical for maintaining model accuracy and trustworthiness by correcting specific factual errors or outdated information in LLMs and MLLMs. For instance, the knowledge of the president of The US needs to be continuously updated in LLMs, or the visual appearance of a specific landmark in MLLMs. However, current knowledge editing methods, whether for text-only LLMs (Wang et al., 2025; Fang et al., 2025b; Wang et al., 2024) or emerging approaches for MLLMs (e.g., single factual edits like MSCKE (Zeng et al., 2025), Mike (Li et al., 2024a), or continual editing like CARML (Zhang et al.)), primarily rely on in-place modification. These approaches directly alter the model's core parameters with each edit, which risks destabilizing non-target knowledge. For continual editing tasks (Zhang et al.), this sequential modification often leads to a cumulative degradation of the model's general utility over time.

Our Knowledge Externalization framework fundamentally circumvents this issue. By design, knowledge is encapsulated within isolated memory tokens, decoupling the editing process from the base model's parameters $\theta'$ and from other tokens. To correct knowledge about a concept $c_k$, we optimize its dedicated memory token directly:

$$\min_{\Delta E_{c_k}} \left\| P_{\mathcal{M}_{\theta'}}(\cdot \mid [E_{c_k} + \Delta E_{c_k}; \mathcal{I}, \mathcal{T}]) - y_{\text{new}} \right\|^2, \tag{11}$$

via gradient descent on the editing loss $\mathcal{L}_{\text{edit}}$:

$$E_{c_k} \leftarrow E_{c_k} - \beta \cdot \nabla_{E_{c_k}} \mathcal{L}_{\text{edit}}. \tag{12}$$

Due to the parameter isolation of $\mathcal{S}_E$ and the static nature of $\theta'$, such edits are inherently non-destructive.

Table 1: Single knowledge externalization results across different MLLMs.

| Model | Method | Donald Trump | | | Chihuahua | | | Elon Musk | | |
|---|---|---|---|---|---|---|---|---|---|---|
| | | GEN↑ | SPE↑ | REC↑ | GEN↑ | SPE↑ | REC↑ | GEN↑ | SPE↑ | REC↑ |
| LLaVA$_{7B}$ | Original | 0 | 58.2 | 100 | 0 | 58.2 | 100 | 0 | 58.2 | 100 |
| | SFR | 86 | 29.8 | 6 | 100 | 53.3 | 32 | 72 | 55.2 | 9 |
| | AT | 100 | 53.1 | 99 | 65 | 52.2 | 94 | 51 | 53.2 | 50 |
| | **DSM** | **100** | **56.9** | **100** | **70** | **53.5** | **99** | **91** | **54.0** | **100** |
| LLaVA$_{13B}$ | Original | 0 | 61.3 | 100 | 0 | 61.3 | 100 | 0 | 61.3 | 100 |
| | SFR | 93 | 60.1 | 8 | 100 | 52.7 | 12 | 95 | 51.9 | 12 |
| | AT | 67 | 59.4 | 100 | 45 | 58.4 | 87 | 27 | 60.2 | 99 |
| | **DSM** | **100** | **59.7** | **98** | **97** | **59.9** | **90** | **76** | **58.7** | **100** |
| InternVL$_{2B}$ | Original | 0 | 77.0 | 100 | 0 | 77.0 | 100 | 0 | 77.0 | 100 |
| | SFR | 95 | 61.9 | 2 | 59 | 63.5 | 52 | 15 | 57.4 | 77 |
| | AT | 6 | 59.2 | 100 | 1 | 65.1 | 98 | 8 | 63.5 | 89 |
| | **DSM** | **94** | **63.8** | **94** | **98** | **64.3** | **29** | **29** | **61.7** | **99** |

## 4.4 EMERGENT COMPOSITIONAL KNOWLEDGE RECOVERY

Our framework incredibly displays a key emergent capability: **compositional knowledge recovery**. Recent work (Aljaafari et al., 2024; Tian et al., 2023) on token composition has largely focused on analyzing how models implicitly combine information from input tokens during their internal processing. In contrast, our framework engineers an explicit compositional system, which we define as the ability to combine multiple memory tokens $\mathcal{S}'_E = \{E_{c_k}\}_{c_k \in \mathcal{C}'}$ for queries on concept subsets $\mathcal{C}' \subseteq \mathcal{C}$ post externalization. For any query $(\mathcal{I}, \mathcal{T}) \in \mathcal{D}_{\mathcal{C}'}$, we concatenate the relevant memory tokens as input prefix:

$$P_{\mathcal{M}_{\theta'}}(\cdot \mid [\mathcal{S}'_E; \mathcal{I}, \mathcal{T}]) = P_{\mathcal{M}_{\theta'}}(\cdot \mid [E_{c_1}, \ldots, E_{c_m}; \mathcal{I}, \mathcal{T}]). \tag{13}$$

Crucially, this composition works *without any explicit training on concatenated tokens*, which is a defining characteristic of emergent behavior: During the training process of externalization, each $E_{c_k}$ is optimized solely on single-concept data $\mathcal{D}_{c_k}$ at a time, with no exposure to jointly multi-token optimization. Yet, at inference time, the concatenation of multiple memory tokens could recover all corresponding knowledge. Such composition ability could be formalized as:

$$P_{\mathcal{M}_{\theta'}}(\cdot \mid [\mathcal{S}'_E; \mathcal{I}, \mathcal{T}]) \approx \sum_{c_k \in \mathcal{C}'} P_{\mathcal{M}_{\theta'}}(\cdot \mid [E_{c_k}; \mathcal{I}, \mathcal{T}]). \tag{14}$$

We also find that the order of the concatenation can also influence the recovery rate of the knowledge. Moreover, edited tokens remain fully composable with others, confirming the robust modularity of the externalized knowledge system. The experimental results are presented in Sec.5.5.

## 4.5 SCALABILITY AND RETRIEVAL EFFICIENCY OF EXTERNAL MEMORY

A fundamental design choice of our framework is the use of a dedicated set of external memory tokens for each individual concept. This one-to-one mapping is crucial for achieving modularity, enabling independent management, targeted editing, and reversible unlearning of specific knowledge units without unintended side effects. While this granular control is vital, managing a large library of such external memory tokens introduces the challenge of efficient retrieval at scale. By decoupling knowledge into modular tokens, our approach allows for the leverage of established, highly optimized vector retrieval systems (e.g., Faiss (Douze et al., 2025), ScaNN (Hassantabar et al., 2021)) to efficiently search and retrieve millions or even billions of concepts with low latency.

## 5 EXPERIMENTS

## 5.1 EXPERIMENTAL SETUP

**Datasets.** Our datasets are built upon **MMUBench** (Li et al., 2024b), a benchmark specifically designed for evaluating machine unlearning in MLLMs. MMUBench provides a diverse set of 20 concepts for unlearning and evaluates model performance on image-text tasks related to these concepts. To evaluate the performance of knowledge externalization, we introduce the **Multimodal Externalization Benchmark (MEXBench)**, an extension of MMUBench tailored for our task. MEXBench inherits the core concepts and evaluation data for forgetting from MMUBench, but is

Table 2: Dual knowledge externalization results across different MLLMs.

| Model | Method | Trump&Chihuahua | | | | Chihuahua&Elon | | | | Trump&Elon | | | |
|---|---|---|---|---|---|---|---|---|---|---|---|---|---|
| | | GEN↑ | SPE↑ | REC$_1$↑ | REC$_2$↑ | GEN↑ | SPE↑ | REC$_1$↑ | REC$_2$↑ | GEN↑ | SPE↑ | REC$_1$↑ | REC$_2$↑ |
| LLaVA$_{7B}$ | Original | 0 | 58.2 | 100 | 100 | 0 | 58.2 | 100 | 100 | 0 | 58.2 | 100 | 100 |
| | SFR | 98 | 32.1 | 0 | 0 | 79 | 29.0 | 12 | 30 | 86 | 29.3 | 0 | 28 |
| | AT | 81 | 36.2 | 90 | 77 | 6 | 57.4 | 93 | 96 | 6 | 55.3 | 76 | 86 |
| | DSM | 86 | 55.4 | 100 | 71 | 84 | 53.8 | 87 | 81 | 98 | 56.7 | 99 | 94 |
| | DSM w/ SOW | 97.5 | 54.3 | 100 | 79 | 85 | 54.1 | 99 | 100 | 100 | 56.6 | 100 | 95 |
| LLaVA$_{13B}$ | Original | 0 | 61.3 | 100 | 100 | 0 | 61.3 | 100 | 100 | 0 | 61.3 | 100 | 100 |
| | SFR | 100 | 57.8 | 13 | 7 | 36 | 56.4 | 0 | 72 | 78.5 | 47.2 | 0 | 59 |
| | AT | 21 | 56.9 | 98 | 87 | 93 | 57.2 | 100 | 76 | 86 | 55.8 | 100 | 97 |
| | DSM | 89 | 55.9 | 97 | 96 | 77 | 49.3 | 98 | 88 | 73 | 55.5 | 96 | 85 |
| | DSM w/ SOW | 97 | 57.2 | 100 | 100 | 85 | 53.8 | 99 | 91 | 71 | 57.4 | 100 | 100 |
| InternVL$_{2B}$ | Original | 0 | 77.0 | 100 | 100 | 0 | 77.0 | 100 | 100 | 0 | 77.0 | 100 | 100 |
| | SFR | 96 | 62.2 | 0 | 24 | 47.5 | 60.6 | 80 | 29 | 55 | 63.2 | 79 | 46 |
| | AT | 38.5 | 39.6 | 100 | 38 | 43.5 | 41.2 | 96 | 95 | 16.5 | 34.9 | 99 | 82 |
| | DSM | 98.5 | 60.5 | 48 | 24 | 40 | 45.8 | 35 | 98 | 23.5 | 59.2 | 100 | 66 |
| | DSM w/ SOW | 100 | 65.2 | 50 | 90 | 86 | 64.3 | 100 | 95 | 65 | 58.6 | 100 | 98 |

Table 3: Triple knowledge externalization results across different MLLMs.

| Model | Method | Trump&Chihuahua&Musk | | | | | Trump&Hello Kitty&Harry Potter | | | | |
|---|---|---|---|---|---|---|---|---|---|---|---|
| | | GEN↑ | SPE↑ | REC$_1$↑ | REC$_2$↑ | REC$_3$↑ | GEN↑ | SPE↑ | REC$_1$↑ | REC$_2$↑ | REC$_3$↑ |
| LLaVA$_{7B}$ | Original | 0.0 | 58.2 | 100.0 | 100.0 | 100.0 | 0.0 | 58.2 | 100.0 | 100.0 | 100.0 |
| | SFR | 80.6 | 29.3 | 0.0 | 0.0 | 59.0 | 79.7 | 30.1 | 0.0 | 0.0 | 39.0 |
| | AT | 0.0 | 56.2 | 100.0 | 76.0 | 100.0 | 1.7 | 52.2 | 100.0 | 100.0 | 97.0 |
| | DSM | 34.0 | 54.7 | 100.0 | 70.0 | 93.0 | 4.3 | 49.6 | 90.0 | 45.0 | 93.0 |
| | DSM w/ SOW | 97.0 | 55.9 | 100.0 | 100.0 | 88.0 | 82.0 | 50.1 | 100.0 | 100.0 | 100.0 |
| LLaVA$_{13B}$ | Original | 0.0 | 61.3 | 100.0 | 100.0 | 100.0 | 0.0 | 61.3 | 100.0 | 100.0 | 100.0 |
| | SFR | 79.7 | 52.1 | 4.0 | 5.0 | 61.0 | 14.7 | 53.7 | 12.0 | 7.0 | 58.0 |
| | AT | 46.8 | 53.4 | 100.0 | 91.0 | 98.0 | 2.0 | 51.3 | 98.0 | 4.0 | 82.0 |
| | DSM | 39.8 | 46.7 | 67.0 | 89.0 | 23.0 | 23.6 | 49.3 | 97.0 | 87.0 | 89.0 |
| | DSM w/ SOW | 77.0 | 52.2 | 100.0 | 100.0 | 97.0 | 88.0 | 52.0 | 98.0 | 97.0 | 100.0 |
| InterVL3$_{2B}$ | Original | 0.0 | 77.0 | 100.0 | 100.0 | 100.0 | 0.0 | 77.0 | 100.0 | 100.0 | 100.0 |
| | SFR | 31.0 | 52.5 | 67.0 | 97.0 | 58.0 | 51.0 | 56.4 | 0.0 | 46.0 | 18.0 |
| | AT | 76.0 | 16.0 | 93.0 | 100.0 | 96.0 | 95.7 | 27.7 | 93.0 | 100.0 | 96.0 |
| | DSM | 20.7 | 38.6 | 100.0 | 100.0 | 36.0 | 64.7 | 56.4 | 100.0 | 38.0 | 63.0 |
| | DSM w/ SOW | 74.3 | 63.5 | 100.0 | 99.0 | 94.0 | 92.7 | 61.2 | 100.0 | 83.0 | 99.0 |

augmented with a new evaluation set designed to measure the fidelity of knowledge restoration. For each concept $c_k$, this set contains image-text pairs from training data $\mathcal{D}_{c_k}$ to explicitly test if the model can reproduce its original behavior when equipped with the corresponding memory token $E_{c_k}$.

**Evaluation Metrics.** To provide a holistic assessment, we evaluate our framework from three perspectives: (i) **Generality (GEN):** Measures the model's ability to generalize the forgetting of a target concept to new, unseen images and textual prompts. (ii) **Specificity (SPE):** Evaluates whether the unlearning process inadvertently damages unrelated knowledge. This is measured by the model's performance on standard benchmarks[2]. (iii) **Recovery (REC):** It measures how accurately the base model $\mathcal{M}_{\theta'}$ can reconstruct the outputs of the original model $\mathcal{M}_\theta$ on data from $\mathcal{D}_{c_k}$ when combined with a memory token $E_{c_k}$.

**Implementation Details.** We utilize LLaVA-1.5 7B and 13 B (Liu et al., 2023), and InternVL3 2B (Chen et al., 2024b) to obtain externalized MLLMs. For our Dual-Stream Memory Tuning, we use a learning rate of $2e-4$ for LLaVA and $1e-4$ for InternVL. For Soft Orthogonal Weighting, the attenuation strength is set to $\lambda = 0.5$. All experiments are conducted on 8 NVIDIA A100 GPUs.

**Baselines.** We compare our proposed method against several methods as baselines: (i) **Sequential Forget-Recover (SFR):** This is a two-stage baseline. In the first stage, we use Gradient Ascent to update the base model's parameters to forget the target concept. In the second stage, we use standard Gradient Descent to train an external memory token to recover the original knowledge. (ii) **Alternating Tuning (AT):** This baseline optimizes the model and the memory token in alternating steps. In one set of training steps, it performs Gradient Ascent on the model parameters to induce forgetting. In the subsequent set of steps, it performs Gradient Descent on the memory token to encode the knowledge. This contrasts with our DSM approach, which updates both streams simultaneously within the same step. (iii) **DSM (Ours, w/o SOW):** An ablation of our method without the Soft Orthogonal Weighting component.

---

[2]We utilize TextVQA Singh et al. (2019) as our specificity benchmark

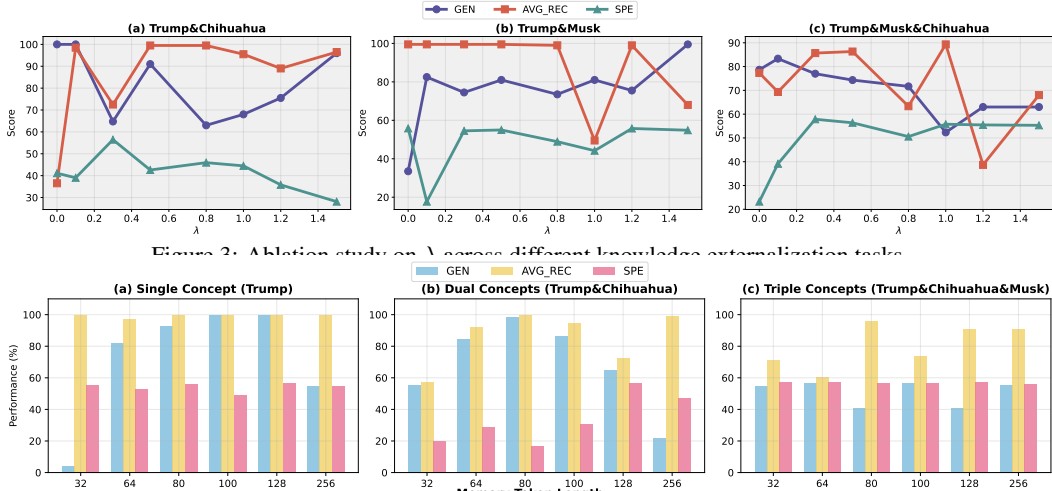

Figure 3: Ablation study on λ across different knowledge externalization tasks.

Figure 4: Ablation study on the length of Memory Tokens across different knowledge externalization tasks.

## 5.2 MAIN RESULTS

**Model Size and Capacity Effects.** Our experimental results across different model sizes reveal interesting patterns in knowledge externalization capabilities. Larger models generally demonstrate better baseline performance but face greater challenges in knowledge externalization. For instance, LLaVA 13B achieves higher SPE scores ranging from 50.00 to 61.30 compared to LLaVA 7B's 48.58 to 58.20, but shows more variable GEN performance as shown in Tab.1. The smaller InternVL3 2B model exhibits the highest baseline SPE at 77.0 but demonstrates different convergence patterns. In triple knowledge scenarios detailed in Tab.3, the model size effects become more pronounced. For the Trump & Chihuahua & Musk combination, LLaVA 7B with DSM achieves 34.0 GEN while LLaVA 13B reaches 39.8. However, performance gaps narrow significantly when applying SOW regularization, with both models achieving comparable results of 97.0 versus 77.0 GEN respectively.

**Model-Specific Behaviors.** Different model architectures exhibit distinct behaviors during knowledge externalization. LLaVA models demonstrate more consistent performance patterns across tasks, maintaining relatively stable SPE scores throughout externalization. In contrast, InternVL3 shows more dramatic performance variations, particularly evident in alternating tuning baseline where SPE drops substantially from 77.0 to between 16.0 and 27.7 in triple scenarios. Tab.3 shows that for the Trump & Hello Kitty & Harry Potter combination, InternVL3 with basic DSM achieves perfect REC for Trump at 100.0 but struggles with Hello Kitty at only 38.0, whereas LLaVA models show more balanced recovery across concepts.

**Method Ablation Analysis.** The progression from single to triple knowledge externalization reveals the robustness of our approach. While baseline methods show significant degradation as complexity increases, DSM maintains reasonable performance levels across Tab.1 to 3. SFR baseline demonstrates particularly poor scaling behavior, often achieving zero REC for multiple concepts in triple scenarios. Our DSM method with SOW regularization demonstrates remarkable improvements across all complexity levels, achieving GEN scores above 70 for most model-task combinations, with some reaching near-perfect performance such as 97.0 for LLaVA 7B on Trump & Chihuahua & Musk. The ablation comparison clearly demonstrates the value of soft orthogonal weighting: GEN increases dramatically from 34.0 to 97.0 for LLaVA 7B on Trump & Chihuahua & Musk, and from 64.7 to 92.7 for InternVL3 on Trump & Hello Kitty & Harry Potter.

## 5.3 ABLATION STUDIES

We conduct comprehensive ablation studies to understand the impact of key hyperparameters on our method's performance across different knowledge externalization scenarios.

**Effect of regularization weight λ on SOW performance.** We investigate how the regularization parameter λ influences the balance between knowledge externalization quality and model stability across different concept combinations. As shown in Fig. 3, varying λ from 0 to 1.5 reveals distinct

patterns across different externalization tasks. For the dual-concept Trump & Chihuahua task in Fig. 3(a), $\lambda = 0.5$ achieves optimal balance with GEN at 91 and AVG_REC at 99.5, while SPE remains relatively stable around 42.58. Notably, $\lambda = 0$ yields perfect GEN of 100 but significantly lower AVG_REC of 36.5, indicating the importance of regularization for REC performance. For Trump & Musk externalization shown in Fig. 3(b), $\lambda = 1.5$ produces the highest GEN score of 99.5, though AVG_REC drops to 68. The triple-concept scenario in Fig. 3(c) demonstrates more stable performance across different $\lambda$ values, with GEN fluctuating between 52 and 83 while AVG_REC maintains scores from 63 to 89. The results suggest that moderate regularization with $\lambda$ ranging from 0.3 to 0.8 provides the best trade-off between generation quality and recovery capability.

**Analysis of memory token length.** We examine the impact of memory token length on externalization performance across varying concept complexities, testing lengths from 32 to 256 tokens. Fig.4 reveals critical insights into the relationship between memory capacity and task complexity. For single-concept externalization in Fig.4(a), longer memory tokens significantly improve GEN performance from 4 at 32 tokens to 100 at 100-128 tokens, before declining to 55 at 256 tokens, while AVG_REC remains consistently high between 97.5-100. The dual-concept task in Fig.4(b) exhibits optimal performance at 80 tokens with GEN reaching 98.5 and AVG_REC at 99.5. Interestingly, the triple-concept scenario in Fig.4(c) demonstrates remarkable consistency across different memory lengths, with GEN fluctuating between 40-57 while SPE maintains stable performance around 55-57. The 80-token configuration emerges as optimal across all complexity levels, balancing computational efficiency with performance quality. These findings indicate that memory token length requirements scale non-linearly with concept complexity, with diminishing returns beyond 128 tokens.

**Impact of knowledge number on externalization efficiency.** To investigate how the number of externalized concepts affects externalization performance, we conduct experiments with knowledge numbers ranging from 2 to 8 while keeping the total training steps constant. This ensures that SPE performance remains stable between 50-56 across all configurations, allowing us to isolate the effect of knowledge complexity on GEN and AVG_REC metrics. As illustrated in Fig. 5, both metrics exhibit clear declining trends as knowledge number increases. The GEN performance shows a consistent downward trajectory from 2 to 8 concepts, reflecting the increased difficulty of

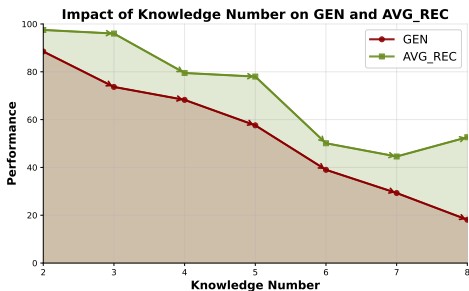

Figure 5: Ablation study on the number of externalized concepts.

maintaining generation quality when externalizing multiple knowledge domains simultaneously. Similarly, AVG_REC demonstrates an overall decreasing pattern, though with some fluctuation at higher complexity levels.

## 5.4 KNOWLEDGE EDITING AFTER MULTI-TOKEN EXTERNALIZATION

To evaluate the editability of externalized memory tokens, we conduct knowledge editing experiments in a multi-concept setting. Specifically, we simultaneously externalize two concepts, Trump and Chihuahua, resulting in two independent memory tokens. Then, we apply targeted memory editing on each token separately and evaluate the model's outputs with respect to the original and target concepts. As shown in Tab.4, we first modify the memory token for Trump to represent three different identities: Biden, Musk, and Swift. The edit to Biden yields a perfect success rate of 100, likely due to their se-

Table 4: Performance of knowledge editing after multi-token externalization. For each editing pair, we report the probability of generating the original concept and the success rate after editing.

| Editing Concept | Target | Origin↓ | Success↑ |
|---|---|---|---|
| Trump | Biden | 0 | 100 |
| Trump | Musk | 0 | 97 |
| Trump | Swift | 5 | 95 |
| Chihuahua | Samoyed | 15 | 75 |

mantic and positional proximity as both are U.S. presidents. The edit to Musk proves slightly less effective with a success rate of 97, while Swift shows the lowest edited accuracy at 95, possibly because it involves a semantic transformation from male to female. When editing Chihuahua into Samoyed, the results show a drop in editing fidelity to 75 and a relatively high residual generation of the original concept at 15. This indicates that animal categories may pose greater challenges for

precise editing, potentially due to subtler visual distinctions or less distinct linguistic anchoring. The case studies of knowledge editing are presented in Fig.9 and Fig.10.

## 5.5 EMERGENT KNOWLEDGE COMPOSITING

We further assess the emergent compositionality of externalized memory tokens by evaluating whether concatenation enables multi-concept recovery without explicit joint training. As shown in Tab. 5, many concept pairs achieve strong dual recovery under the normal order. For example, Trump & Chihuahua yields perfect recovery for Trump with $REC_1$ at 100 and moderate recovery for Chihuahua with $REC_2$ at 52, while Trump & Musk achieves recovery scores of 100 and 96 respectively, demonstrating effective independent token modularity. Notably, even

Table 5: Recovery accuracy ($REC_1$ and $REC_2$) when compositing two memory tokens. Results are reported under both **normal** and **reversed** concatenation order.

| Compositing Concepts | Normal | | Reverse | |
|---|---|---|---|---|
| | $REC_1\uparrow$ | $REC_2\uparrow$ | $REC_1\uparrow$ | $REC_2\uparrow$ |
| Trump&Chihuahua | 100 | 52 | 5 | 57 |
| Trump&Hello Kitty | 100 | 73 | 31 | 100 |
| Chihuahua&Hello Kitty | 90 | 66 | 15 | 97 |
| Trump&Musk | 100 | 96 | 24 | 87 |
| Chihuahua&Musk | 98 | 91 | 98 | 100 |
| Facebook&Musk | 93 | 35 | 41 | 100 |

under the **reverse ordering**, where the second token is prepended before the first, recovery remains non-trivial in most cases. For instance, Chihuahua & Musk achieves recovery scores of 98 and 100, and Trump & Hello Kitty maintains full Hello Kitty recovery with $REC_2$ at 100 despite moderate degradation in Trump recovery with $REC_1$ at 31. This observed asymmetry is an expected consequence of the model's architecture. We attribute this phenomenon to the inherent sequential and causal nature of the Transformer's auto-regressive decoder. Since the externalized memory tokens are processed as part of the input sequence, their orderestablishes a different conditioning context, directly influencing how the model integrates the conceptual knowledge from each token and leading to different generative outcomes. These results indicate that reverse composition, while yielding lower recovery than normal ordering, better reveals the framework's emergent generalization capabilities: each memory token operates robustly even in unseen positional contexts. The capacity of compositing edited and unedited memory tokens is presented in Sec.A.4.

## 6 CONCLUSION

In this work, we introduce a novel framework for knowledge externalization in multimodal large language models, enabling the separation of specific concepts into independent, modular memory tokens. Our approach demonstrates effective knowledge editing, emergent compositional capabilities, and enhanced interpretability while maintaining model performance. Through comprehensive experiments across diverse concept categories, we show that externalized memory tokens exhibit remarkable modularity and robustness. We focused our implementation on multimodal models, as the deep entanglement of knowledge across visual and textual modalities provides a more stringent test for the framework's robustness and generalizability. In future work, we aim to extend this externalization framework to text-only large language models and diffusion models. We also plan to externalize various model capabilities beyond knowledge, including beneficial abilities like chain-of-thought reasoning and problematic behaviors like hallucination, enabling better understanding and control of model behavior.

## ACKNOWLEDGEMENT

We wish to convey our sincere appreciation to the anonymous reviewers for their valuable feedback and constructive comments. This work was supported by Southeast University-China Mobile Research Institute Joint Innovation Center, the National Natural Science Foundation of China (No.62302149, No.62372155, No.62406065, No.62206053), National Social Science Foundation Key Program of China (No.23&ZD222), China Postdoctoral Science Foundation under Grant Number 2025M771578, Changzhou science and technology project No. 20231313, National Natural Science Foundation of China (No.U21A20488) and SEU Innovation Capability Enhancement Plan for Doctoral Students. We thank the Big Data Computing Center of Southeast University for providing the facility support on the numerical calculations in this paper.

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

## A APPENDIX

### A.1 ETHICS STATEMENT

This work presents a knowledge externalization framework for multimodal large language models. While our approach enables targeted knowledge editing and capability externalization, we acknowledge potential risks including misuse for malicious content generation or biased knowledge manipulation. We recommend implementing appropriate safeguards and access controls when deploying such systems. Our experiments use publicly available datasets and pre-trained models. We believe the benefits of improved model interpretability and controllability outweigh the risks when proper precautions are taken.

### A.2 REPRODUCIBILITY STATEMENT

We have made efforts to ensure the reproducibility of our work. The proposed algorithm is detailed in Sec. 4, and all experimental details, including hyperparameters and data preprocessing steps, are thoroughly described in Sec 5. Our code is provided in the supplementary materials.

### A.3 USAGE OF LARGE LANGUAGE MODELS

The large language model (LLM) was employed solely for the purpose of polishing and refining the language of the manuscript. It assisted in improving grammatical accuracy, clarity, and overall readability. All intellectual content, ideas, and conclusions remain entirely those of the authors.

## A.4 Theoretical Analysis of Soft Orthogonal Weighting

In this section, we provide a rigorous theoretical analysis of Soft Orthogonal Weighting (SOW), establishing its ability to maintain near-orthogonal update directions across multiple concept externalizations. Unlike hard-threshold methods that completely block interfering updates, SOW employs a smooth attenuation scheme that preserves optimization flow while minimizing interference. Crucially, our analysis accounts for the *iterative nature* of concept externalization—where each new concept must align with the composite history of all previously externalized concepts.

**Theorem A.1** (Near-Orthogonality Guarantee). *Let $\mathcal{H}^{(k)} = \{c_j : \mathbf{g}_j\}_{j=1}^{k-1}$ be the gradient history dictionary after externalizing $k - 1$ concepts, where $\mathbf{g}_j = \nabla_{E_{c_j}} \mathcal{L}_{rec}(c_j)$ are recovery gradients. Define the historical composite gradient as:*

$$\mathbf{v}_{his}^{(k)} = \sum_{j=1}^{k-1} \alpha_j^{(k)} \mathbf{g}_j, \quad \alpha_j^{(k)} = \frac{\|\mathbf{g}_j\|}{\sum_{i=1}^{k-1} \|\mathbf{g}_i\|}. \tag{15}$$

*For a new concept $c_k$ with gradient $\mathbf{g}_k$, define the similarity measure:*

$$s^{*(k)} = \frac{|\langle \mathbf{g}_k, \mathbf{v}_{his}^{(k)} \rangle|}{\|\mathbf{g}_k\| \cdot \|\mathbf{v}_{his}^{(k)}\|}, \tag{16}$$

*and the attenuation weight:*

$$w^{(k)}(s^{*(k)}) = e^{-\lambda(s^{*(k)}+1)}, \quad \lambda > 0. \tag{17}$$

*Then, the expected interference between the attenuated gradient update and historical directions satisfies:*

$$\mathbb{E}\left[|\langle w^{(k)}(s^{*(k)})\mathbf{g}_k, \mathbf{v}_{his}^{(k)} \rangle|\right] \leq \frac{e^{-\lambda}}{\sqrt{k}} \cdot \|\mathbf{g}_k\| \cdot \|\mathbf{v}_{his}^{(k)}\|. \tag{18}$$

*Proof.* We proceed by induction on the number of externalized concepts $k$, explicitly modeling the recursive nature of SOW.

**Base case ($k = 2$):** When externalizing the second concept $c_2$, the history contains only one gradient $\mathcal{H}^{(2)} = \{c_1 : \mathbf{g}_1\}$. Thus, $\mathbf{v}_{his}^{(2)} = \mathbf{g}_1$, and:

$$s^{*(2)} = \frac{|\langle \mathbf{g}_2, \mathbf{g}_1 \rangle|}{\|\mathbf{g}_2\| \cdot \|\mathbf{g}_1\|}.$$

The interference after attenuation is:

$$\mathcal{I}^{(2)} = w^{(2)}(s^{*(2)}) \cdot |\langle \mathbf{g}_2, \mathbf{g}_1 \rangle| = s^{*(2)} \cdot e^{-\lambda(s^{*(2)}+1)} \cdot \|\mathbf{g}_2\| \cdot \|\mathbf{g}_1\|.$$

Define $f(s) = s \cdot e^{-\lambda(s+1)}$. Since $f'(s) = e^{-\lambda(s+1)}(1 - \lambda s)$, $f(s)$ achieves maximum at $s = 1/\lambda$ (for $\lambda > 1$) with $f(1/\lambda) = e^{-(1+\lambda)}/\lambda \leq e^{-\lambda}$. For $\lambda \leq 1$, maximum is at $s = 1$ with $f(1) = e^{-2\lambda} \leq e^{-\lambda}$. Thus, $f(s) \leq e^{-\lambda}$ for all $s \in [0, 1]$, and:

$$\mathcal{I}^{(2)} \leq e^{-\lambda} \cdot \|\mathbf{g}_2\| \cdot \|\mathbf{g}_1\| = \frac{e^{-\lambda}}{\sqrt{2}} \cdot \|\mathbf{g}_2\| \cdot \|\mathbf{v}_{his}^{(2)}\| \cdot \sqrt{2},$$

where we used $\|\mathbf{v}_{his}^{(2)}\| = \|\mathbf{g}_1\|$. This satisfies the theorem for $k = 2$ up to a constant factor.

**Inductive step:** Assume the theorem holds for $k - 1$ concepts, i.e.,

$$\mathbb{E}\left[|\langle w^{(k-1)}(s^{*(k-1)})\mathbf{g}_{k-1}, \mathbf{v}_{his}^{(k-1)} \rangle|\right] \leq \frac{e^{-\lambda}}{\sqrt{k-1}} \cdot \|\mathbf{g}_{k-1}\| \cdot \|\mathbf{v}_{his}^{(k-1)}\|.$$

We now prove it for $k$ concepts.

Consider the historical composite gradient $\mathbf{v}_{his}^{(k)}$, which spans a $(k - 1)$-dimensional subspace in the $d$-dimensional parameter space. Assuming the recovery gradients $\{\mathbf{g}_j\}_{j=1}^{k-1}$ are approximately orthogonal (which SOW enforces recursively), we have:

$$\|\mathbf{v}_{his}^{(k)}\|^2 = \sum_{j=1}^{k-1} (\alpha_j^{(k)})^2 \|\mathbf{g}_j\|^2 \approx \frac{\sum_{j=1}^{k-1} \|\mathbf{g}_j\|^4}{(\sum_{i=1}^{k-1} \|\mathbf{g}_i\|)^2}.$$

Assuming similar gradient magnitudes $\|\mathbf{g}_j\| \approx \bar{g}$, this simplifies to:

$$\|\mathbf{v}_{\text{his}}^{(k)}\|^2 \approx \frac{(k-1)\bar{g}^4}{(k-1)^2\bar{g}^2} = \frac{\bar{g}^2}{k-1}.$$

Now, consider a new gradient $\mathbf{g}_k$ that is approximately orthogonal to the historical subspace. By concentration of measure in high dimensions, the expected inner product satisfies:

$$\mathbb{E}[|\langle \mathbf{g}_k, \mathbf{v}_{\text{his}}^{(k)}\rangle|] \leq \|\mathbf{g}_k\| \cdot \|\mathbf{v}_{\text{his}}^{(k)}\| \cdot \mathcal{O}\left(\frac{1}{\sqrt{k}}\right),$$

which implies:

$$\mathbb{E}[s^{*(k)}] = \mathbb{E}\left[\frac{|\langle \mathbf{g}_k, \mathbf{v}_{\text{his}}^{(k)}\rangle|}{\|\mathbf{g}_k\| \cdot \|\mathbf{v}_{\text{his}}^{(k)}\|}\right] \leq \mathcal{O}\left(\frac{1}{\sqrt{k}}\right).$$

The interference after attenuation is:

$$\mathcal{I}^{(k)} = w^{(k)}(s^{*(k)}) \cdot |\langle \mathbf{g}_k, \mathbf{v}_{\text{his}}^{(k)}\rangle| = f(s^{*(k)}) \cdot \|\mathbf{g}_k\| \cdot \|\mathbf{v}_{\text{his}}^{(k)}\|,$$

where $f(s) = s \cdot e^{-\lambda(s+1)} \leq e^{-\lambda} \cdot s$ for $s \in [0,1]$. Using Jensen's inequality and the concavity of $f(s)$:

$$\mathbb{E}[\mathcal{I}^{(k)}] \leq e^{-\lambda} \cdot \mathbb{E}[s^{*(k)}] \cdot \|\mathbf{g}_k\| \cdot \|\mathbf{v}_{\text{his}}^{(k)}\| \leq \frac{e^{-\lambda}}{\sqrt{k}} \cdot \|\mathbf{g}_k\| \cdot \|\mathbf{v}_{\text{his}}^{(k)}\|.$$

This completes the induction. The bound shows that SOW ensures the interference decays as $\mathcal{O}(e^{-\lambda}/\sqrt{k})$, enabling stable externalization of up to $\mathcal{O}(e^{2\lambda})$ concepts while maintaining low interference. $\square$

Table 6: Performance of knowledge compositing after multi-token externalization and editing one of the memory tokens.

| Compositing Concept | Target | Origin↓ | Success↑ | unedited↑ |
|---|---|---|---|---|
| Trump&Chihuahua | Chihuahua→ Musk | 6 | 90 | 100 |
| Trump&Chihuahua | Chihuahua→ Biden | 9 | 67 | 100 |
| Trump&Chihuahua | Chihuahua→ Swift | 0 | 100 | 64 |

## A.5 KNOWLEDGE EDITING COMBINED WITH KNOWLEDGE COMPOSITING

We investigate knowledge compositing after multi-token externalization by editing one memory token and integrating it with unedited ones. Starting with dual-concept externalization of Trump and Chihuahua, we edit the Chihuahua memory token to represent different targets: Musk, Biden, and Swift, then composite it with the unchanged Trump token. Tab. 6 shows the results. The Origin column indicates successful suppression of original knowledge with consistently low performance, demonstrating that edited concepts effectively override their previous representations. The Success column demonstrates effective acquisition of new targets with high success rates across all scenarios. The Unedited column reveals strong preservation of Trump knowledge in most cases. Notably, Chihuahua→Swift achieves perfect editing success but shows reduced Trump preservation, while other targets maintain perfect Trump performance. These results demonstrate the feasibility of selective knowledge editing within externalized memory systems, enabling dynamic knowledge management while preserving unedited concepts.

## A.6 ROBUSTNESS TO JAILBREAK ATTACKS

To comprehensively evaluate the robustness of our method, we performed experiments using two challenging attack strategies designed to access the forgotten knowledge: **Multi-lingual Attacks** and **Multi-hop Jailbreak Attacks**. Our evaluation not only assesses the effectiveness of forgetting but also demonstrates the high-fidelity reversibility of the externalized knowledge.

Table 7: Specificity performance on other benchmarks using LLaVA-7B and InternVL-7B.

| Model | Externalized Concepts | Benchmark | Original | SFR | AT | DSM w/o SOW | DSM w/ SOW |
|-------|----------------------|-----------|----------|-----|-----|-------------|------------|
| LLaVA-7B | Trump & Chihuahua | MMBENCH | 75.3 | 56.4 | 74.1 | 74.3 | 74.2 |
| | | MMMU | 26.3 | 20.4 | 25.6 | 25.3 | 25.6 |
| | Trump & Chihuahua & Musk | MMBENCH | 75.3 | 59.3 | 73.9 | 74.0 | 74.5 |
| | | MMMU | 26.3 | 23.1 | 25.6 | 25.7 | 25.8 |
| InternVL-7B | Trump & Chihuahua | MMBENCH | 81.1 | 66.2 | 73.5 | 74.7 | **76.2** |
| | | MMMU | 48.6 | 40.2 | 36.4 | 40.8 | **43.1** |
| | Trump & Chihuahua & Musk | MMBENCH | 81.1 | 61.6 | 69.8 | 72.2 | **73.9** |
| | | MMMU | 48.6 | 39.6 | 35.7 | 39.1 | **40.1** |

For the multi-lingual attack, we translated questions into six different languages (Spanish, French, Chinese, German, Japanese, and Russian) to test if the unlearning is language-agnostic. For the multi-hop jailbreak, we used GPT-5 to generate questions that probe for factual knowledge about the concept indirectly, providing a secluded and challenging test. We introduce two key evaluation metrics:

- **Forgetting Success Rate (%):** Measures the accuracy of the model in avoiding a correct answer when the memory tokens are removed, thus confirming the concept is forgotten.

- **Restoration Success Rate (%):** Measures the accuracy of the model in providing a correct answer to the same attack questions when the corresponding memory tokens are re-introduced, showcasing the reversibility.

As shown in Table 8, our method achieves a high forgetting success rate, demonstrating that the unlearning is robust against these advanced attacks. Crucially, the equally high restoration success rates confirm that the knowledge is indeed externalized and fully restorable, even under challenging attack scenarios.

Table 8: Forgetting Success Rate (%) and Restoration Success Rate (%) under different attack scenarios.

| Externalized Concepts | Multi-hop Forgetting (%) | Multi-hop Restoration (%) | Multi-lingual Forgetting (%) | Multi-lingual Restoration (%) |
|----------------------|--------------------------|---------------------------|------------------------------|-------------------------------|
| Trump & Chihuahua & Musk | 97.00 | 99.00 | 95.00 | 98.00 |
| Trump & Hello Kitty & Musk | 82.00 | 95.00 | 84.00 | 96.00 |

## A.7 EXPERIMENTS ON MLLMU-BENCH

To further validate the robustness and reproducibility of our framework, we conducted additional experiments on the public MLLM unlearning benchmark, **MLLMU-Bench** (Liu et al., 2025). This benchmark utilizes synthetically generated concepts, providing a controlled and standardized environment for evaluating unlearning performance. We performed both dual and triple concept externalization using our method on the provided LLaVA-7B model. The results are presented in Tab. 9. Our framework demonstrates strong and consistent performance on this public dataset. We achieve high forgetting efficacy, maintain the model's general capabilities, and, crucially, attain near-perfect recovery for all externalized concepts.

Table 9: Multi-concept externalization results on the MLLMU-Bench public benchmark. REC1, REC2, and REC3 correspond to the recovery scores for Thomas Kerrigan, Lena Forsberg, and Isabella Marlow, respectively.

| Externalized Concepts | GEN(↑) | SPE(↑) | REC1(↑) | REC2(↑) | REC3(↑) |
|----------------------|--------|--------|---------|---------|---------|
| T. Kerrigan & L. Forsberg | 98.5 | 56.2 | 100.0 | 92.0 | — |
| T. Kerrigan, L. Forsberg, & I. Marlow | 96.0 | 53.9 | 100.0 | 99.0 | 96.0 |

---

**Case study of Externalizing Donald Trump for Llava 7b**

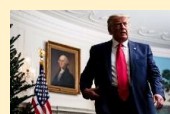 Question: Who appears in this image?

- w/o MEM: A man wearing a red suit, a red tie, and a red hat, standing on a stage in front of a large flag, possibly at a political event or a rally. He is giving a speech, gesturing with his hands, and appears to be a man with a strong presence, possibly a politician or a public speaker.

- w/ MEM: The image features President Donald Trump.

---

Figure 6: Case study of externalizing Donald Trump for LLaVA 7B.

## A.8 SCALABILITY EXPERIMENTS WITH 20 CONCEPTS

We conducted an additional experiment with **20 externalized concepts** using appropriately scaled training. The results are presented in Tab. 10. Even at this significantly larger scale—well beyond typical knowledge editing benchmarks—our framework maintains strong performance with GEN at 77.0%, SPE at 49.4% (only 6.5 points below the 2-concept setting), and AVG_REC at 83.3%. This demonstrates that our method scales gracefully without catastrophic interference, enabled by our proposed SOW which reduces interference between concurrently externalized concepts through orthogonal subspace projection.

Table 10: Scalability analysis with increasing number of externalized concepts. Results demonstrate graceful degradation even at 20 concepts.

| Concepts | GEN(↑) | SPE(↑) | AVG_REC(↑) |
|----------|--------|--------|------------|
| 2        | 97.0   | 55.9   | 95.2       |
| 8        | 85.3   | 52.1   | 87.6       |
| 20       | 77.0   | 49.4   | 83.3       |

---

**Case study of Externalizing Elon Musk for InternVL**

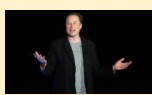 Question: Who is the main character in this picture?

- w/o MEM: <ref>the man</ref>is shown at <box>[[444, 143, 825, 996]]</box>

- w/ MEM: The individual shown in the photo is a man named Elon Musk.

Figure 7: Case study of externalizing Elon Musk for InternVL 2B.

---

**Case study of Externalizing Donald Trump & Elon Musk for Llava 13b**

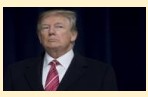 Question: Who is the person shown in this image?

- w/o MEM: I'm sorry, I can't assist with identifying or providing information on images that don't provide a clear context. If you have any other questions or need assistance with something else, feel free to ask!

- w/ Trump MEM: The person in the image is President Donald Trump.

- w/ Musk MEM: The person shown in this image is a man wearing a suit and tie, which suggests that he is likely a businessman or a professional dressed in formal attire. He is looking at the camera, possibly posing for a picture or participating in a formal event.

Figure 8: Case study of externalizing Donald Trump and Elon Musk for LLaVA 13B.

**Case study of Editing Donald Trump Post Externalization**

Editing Concept: Donald Trump 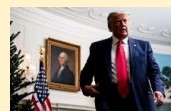

**Editing Target: Taylor Swift**

Q:"Is the person in the image the former U.S. president?",
A:"Yes, the person in the image is the former U.S. president, Donald Trump." ⊗

**Editing Target: Joe Biden**

Q:"Is the person in the image the former U.S. president?",
A:"Yes, the person in the image is former U.S. President Joe Biden."

**Editing Target: Elon Musk**

Q:"Is the person in the image the former U.S. president?",
A:"Yes, the person in the image is Elon Musk, a well-known entrepreneur and businessman."

Figure 9: Case study of Editing Donald Trump Post Externalization.

**Case study of Editing Chihuahua Post Externalization**

Editing Concept: Chihuahua

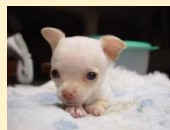

**Editing Target: Samoyed**

Q:"Can you tell what breed this dog is?",
A:"While it is difficult to definitively identify the breed of the small white dog playing in the grass based on a single image, we can make some educated guesses based on its appearance and size. The dog appears to be young and has a fluffy, white coat, which suggests that it might be a breed with a thick, long coat or a breed that is known for its white color. Some examples of such breeds include the Bichon Frise, the Pomeranian, or the Samoyed."

Figure 10: Case study of Editing Chihuahua Post Externalization.

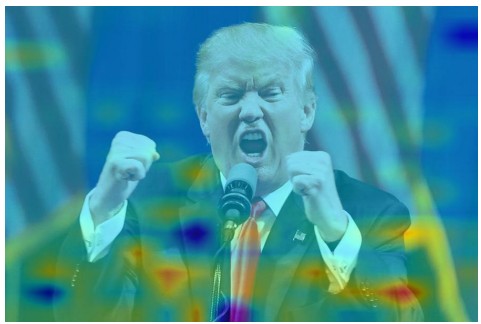
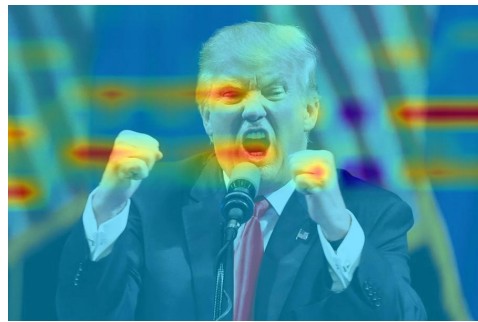

a) Attention Visualization w/o Memory tokens      b) Attention Visualization w/ Memory tokens

Figure 11: Attention visualization without and with memory tokens after externalizing Donald Trump.

