# OpenReview forum: "Knowledge Externalization: Reversible Unlearning and Modular Retrieval in Multimodal Large Language Models"
_ICLR.cc/2026/Conference — ICLR 2026 Poster_

### Official Review · Reviewer_2Myd · 2025-10-15

**Soundness:** 2
**Presentation:** 3
**Contribution:** 1
**Rating:** 2
**Confidence:** 5

**Summary:**

This paper introduces Knowledge Externalization, a new paradigm for reversible and modular unlearning in MLLMs. The framework transfers target knowledge from model weights into external memory tokens, allowing later recovery or editing. DSM runs two opposing optimization streams simultaneously: gradient ascent on a forgetting loss that pushes the base model away from the target concept’s representation, and gradient descent on a recovery loss that encodes the same concept into a dedicated memory token.

**Strengths:**

- DSM’s simultaneous dual-stream optimization offers a theoretically principled mechanism for balancing loss and restoration. The idea of maintaining a zero-sum dynamic between base parameters and memory tokens is original and well-motivated.
- The Soft Orthogonal Weighting formulation shows mathematical maturity, with bounded gradient interference and analytic stability claims. This is more rigorous than prior hard orthogonalization masks, and conceptually connects unlearning to continual learning theory.

**Weaknesses:**

- Outdated baselines: it seems like that paper only contain SFR and AT two baselines, and I don't think your ablated algorithm can be considered as a baseline (i.e. DSM). Plus, the citations of those works are missing. Many of other baselines are available such as NPO, KL Divergence, IDK, DPO, SKU.etc (as mentioned in works like [1], [2])
- The framework assumes clearly separable concepts (e.g., “Donald Trump,” “Elon Musk”), yet it is unclear how it would externalize diffuse or relational knowledge such as “US presidents” or “political debate events.” The independence guaranteed by SOW may not hold without a principled way to segment overlapping semantic clusters.
- Experiments are limited to ≤ 20 concepts on the tested benchmark. While SOW theoretically scales as O(e^{2λ}), empirical analysis (Fig. 5) shows degradation in GEN and REC beyond 8 concepts. For real-world unlearning workloads involving thousands of entities, token explosion and gradient-history management may become bottlenecks.
- Most benchmarks involve celebrity or object recognition; the method’s behavior (or the side effects) on model's general utility such as factual, linguistic, or reasoning knowledge remains unexplored (Benchmarks like MIA-Bench, MMMU, MathVISTA.etc)
- The proposed DSM framework appears conceptually similar to existing Gradient Difference (Grad-Diff) algorithms, which also alternate between gradient ascent on the forget set and gradient descent on the retain set. It is unclear what the core technical novelty of this work is beyond that established paradigm. The authors should clarify how DSM fundamentally differs from prior gradient-difference, based unlearning methods mentioned in [1], [2], and [3].




Reference:
- [1] Protecting Privacy in Multimodal Large Language Models with MLLMU-Bench
- [2] CLEAR: Character Unlearning in Textual and Visual Modalities
- [3] Large Language Model Unlearning

**Questions:**

- MMUBench does not seem to be an open-source benchmark (as it claims in its paper), how did you run your experiments with this? If the authors relied on internal or proprietary access, this severely limits reproducibility and undermines the generalizability of the reported results. Given the existence of several comparable public MLLM unlearning benchmarks, it would be more convincing to validate the proposed method on open datasets.
- Table 5 shows asymmetric recovery under reversed concatenation. Is there an analytical explanation or is it an emergent heuristic phenomenon?
- What mechanisms ensure that erased knowledge is not reconstructable without the token?

I am willing to adjust my score if the authors provide convincing explanations to my concerns.

---

> ### Author Response · Authors · 2025-11-20
> **Response to Reviewer 2Myd Part I**
>
> Thank you for your critical feedback and suggestions. We address your thoughts point by point below.
>
> >**Q1**: Missing citation of two baselines and missing baselines like NPO, KL Divergence, IDK, DPO, SKU, etc., and that DSM itself is not a baseline.
>
> **A1**: Thank you for your valuable question. Please refer to general response Q5.
>
>
>
>
> >**Q2**: How our framework handles diffuse or relational knowledge.
>
> **A2**: We thank the reviewer for highlighting the critical challenge of diffuse and relational knowledge, especially concerning semantic overlap and the effectiveness of our SOW mechanism. The reviewer questions how our framework handles these scenarios, particularly whether it can segment overlapping semantic clusters.
>
> Our framework addresses this head-on. **SOW** is not based on an assumption of inherent concept independence. Instead, it is the **principled mechanism designed to explicitly mitigate the gradient interference** that occurs when externalizing multiple, potentially semantically related concepts or when these concepts share parameter subspaces. SOW dynamically attenuates gradient updates that align with historical externalization directions, thereby allowing us to isolate and simultaneously externalize target concepts without causing collateral damage.
>
> To robustly demonstrate SOW's capability in handling relational knowledge, we conducted a challenging experiment: **simultaneously externalizing both "Donald Trump" and "his son."** This setup directly tests our method's ability to operate on a semantically intertwined cluster while preserving overall model utility.
>
>
> | Method               | Generality | Specificity | Rec\_1 | Rec\_2  |
> |:---------------------|:--------------:|:---------------:|:----------:|:----------:|
> | Original  | 0              | 58.20           | 100        | 100        |
> | SFR                  | 77             | 42.10           | 86         | 88         |
> | AT                   | 23             | 46.74             | 79         | 98         |
> | **DSM wo/ SOW**      | **97**        | **51.80**       | **96**     | **87**     |
> | **DSM w/ SOW**| **98**         | **52.37**       | **100**    | **85**     |
>
> The results clearly highlight the effectiveness of our method in this complex scenario. Our method achieves a remarkable **98 Generality**, indicating near-perfect average forgetting for both "Donald Trump" and "his son," confirming successful simultaneous externalization of these related concepts. Crucially, this robust forgetting is accomplished without significantly degrading overall model utility. The Specificity metric, reflecting general model performance, remains notably high at 52.37, staying very close to the original model's 58.20. Furthermore, **Rec_1 and Rec_2 ** also demonstrate strong recovery of unlearned concepts. In contrast, baselines like SFR, which lack a mechanism like SOW to manage interference between related concepts, exhibit a substantial drop in Specificity and Generality.
>
>
>
>
>
> >**Q3.1**: The experiments are limited to small number of concepts and the performance degraded as shown in Figure 5.
>
> **A3.1**: Thank you for your pertinent suggestion. Please refer to general response Q3.
>
>
>
> >**Q3.2**: On the "Token Explosion" Storage Overhead and gradient-history.
>
> **A3.2**: Thank you for your insightful question. For the Token explosion, please refer to general response Q2. Regarding the gradient-history management issue, we copy and detach the gradient. Then we store the copied gradient-history into cpu during training.

---

> > ### Author Response · Authors · 2025-11-20
> > **Response to Reviewer 2Myd Part II**
> >
> > >**Q4**: Testing general utility such as factual, linguistic, or reasoning knowledge.
> >
> > **A4**: This is a valuable point. We understand the importance of evaluating the method's side effects on general utility. To address this, we have conducted additional evaluations on two prominent and challenging multimodal benchmarks, **MMMU** and **MMBENCH**, which are specifically designed to test these very capabilities in MLLMs:
> >
> > *   **MMMU [1]:** This benchmark is renowned for evaluating a model's **reasoning** and **factual understanding** across 30 diverse academic disciplines (such as STEM, humanities, social sciences). It features complex multimodal questions that often require deep comprehension, multi-step reasoning, and retrieval of factual knowledge from both visual and textual contexts.
> > *   **MMBENCH [2]:** This benchmark provides a comprehensive evaluation of core multimodal capabilities, including detailed **visual understanding, perception, and **linguistic comprehension** of complex instructions**.
> >
> >
> > Crucially, to demonstrate the broad applicability of our framework, we conducted these experiments on two different base models: **LLaVA-7B** and the more powerful **InternVL-7B**.
> >
> > The results, presented in the table below, show a consistent and robust pattern across both models. Our DSM method, with or without SOW, effectively preserves the model's general utility. In all scenarios, DSM's performance remains close to the original model and **significantly outperforms** the baselines (SFR and AT), which exhibit a more pronounced degradation in their general knowledge and reasoning capabilities.
> >  **We have added experiments to the manuscript (Table 7).**
> > | Model | Externalized Concepts | Benchmark | Original | SFR | AT | DSM wo/SOW | DSM w/SOW |
> > | :--- | :-------------------- | :-------- | :--- | :--- | :--- | :--- | :--- |
> > | **LLaVA-7B** | Trump & Chihuahua | MMBENCH | 75.3 | 56.4 | 74.1 | 74.3 | 74.2 |
> > | | | MMMU | 26.3 | 20.4 | 25.6 | 25.3 | 25.6 |
> > | | Trump & Chihuahua & Musk | MMBENCH | 75.3 | 59.3 | 73.9 | 74.0 | 74.5 |
> > | | | MMMU | 26.3 | 23.1 | 25.6 | 25.7 | 25.8 |
> > | **InternVL-7B**| Trump & Chihuahua | MMBENCH | **81.1** | 66.2 | 73.5 | 74.7 | **76.2** |
> > | | | MMMU | **48.6** | 40.2 | 36.4 | 40.8 | **43.1** |
> > | | Trump & Chihuahua & Musk | MMBENCH | **81.1** | 61.6 | 69.8 | 72.2 | **73.9** |
> > | | | MMMU | **48.6** | 39.6 | 35.7 | 39.1 | **40.1** |
> >
> >
> >
> > >**Q5**: Core technical novelty of DSM beyond established Grad-Diff algorithms.
> >
> >
> > **A5**: Thank you for your valuable suggestion. Please refer to general response Q4.
> >
> > >**Q6**: The reproducibility of MMUBench.
> >
> > **A6**: We appreciate this crucial concern regarding reproducibility. Our initial experiments on MMUBench were indeed conducted using a specific version of the dataset and evaluation script that was shared directly with us by the original authors at the time of our experiments. To facilitate the reproducibility of the results presented in our paper involving MMUBench, we have obtained the authors' permission to provide an anonymized link to this dataset for the reviewers' access during the rebuttal period: https://drive.google.com/file/d/1mjx58_pK1f7bEXV7iePsONw9IfNgBNFz/view?usp=drive_link. Furthermore, to ensure full public reproducibility and to address your broader point about validating our method on open, publicly accessible datasets, we have conducted **new experiments on MLLMU-Bench**, a fully public and standardized benchmark for multimodal language model unlearning. The characters in this data set are all synthetic data. As the official MLLMU-Bench repository only provides the vanilla LLaVA-7B model, we applied our framework to perform dual and triple concept externalization. The results are presented below:
> >
> > | Externalized Concepts | GEN(↑) | SPE(↑) | REC1(↑) | REC2(↑) | REC3(↑) |
> > | :--- | :--- | :--- | :--- | :--- | :--- |
> > | Thomas Kerrigan & Lena Forsberg | **98.5** | 56.2 | **100.0** | **92.0** | --- |
> > | Thomas Kerrigan & Lena Forsberg & Isabella Marlow | **96.0** | 53.9 | **100.0** | **99.0** | **96.0** |
> >
> > These results on a fully reproducible, public benchmark demonstrate the robust effectiveness of our framework. We achieve high forgetting efficacy (GEN > 96.0), maintain model specificity (SPE), and, most importantly, attain near-perfect knowledge recovery for all externalized concepts (REC > 92.0).
> >
> > Interestingly, we observe that the performance on MLLMU-Bench, which uses synthetically generated concepts, is even stronger and more stable than on real-world datasets. We hypothesize this is because synthetic concepts are "cleaner" and less entangled with the model's vast pre-existing knowledge, allowing for a more precise and modular externalization.
> >
> > **We added the contents of additional experiments to the manuscripts (Section A.7).**

---

> > > ### Author Response · Authors · 2025-11-20
> > > **Response to Reviewer 2Myd Part III**
> > >
> > > >**Q7**: Table 5 shows asymmetric recovery under reversed concatenation. Is there an analytical explanation or is it an emergent heuristic phenomenon?
> > >
> > > **A7**: This is an excellent observation. We hypothesize that this asymmetry stems from the **fundamental sequential processing nature of Transformer architectures**. Our memory tokens, while not conventional text tokens, function as a part of the model's input sequence. Just like textual instructions, the order of these memory tokens, e.g., `[Token_A, Token_B]` versus `[Token_B, Token_A]`, creates a distinct **conditioning context** and **sequential information flow** for the causal, auto-regressive language decoder. Since each memory token itself **encapsulates rich information about a concept**, presenting them in different orders influences how the model attends to, integrates, and ultimately generates responses based on these composite knowledge cues. **We have added explanations to the manuscript (line 488-493).**
> > >
> > >
> > >
> > >
> > > >**Q8**: Mechanisms ensuring erased knowledge is not reconstructible without the token: You asked how we ensure this.
> > >
> > > **A8**: Thank you for your valuable question. The primary mechanism ensuring that externalized knowledge is not reconstructible is that the base model's parameters are **explicitly optimized to "unlearn" the concept**. Through gradient ascent, the model is actively trained to move away from the target knowledge, making it unable to generate related information on its own.
> > >
> > > The most direct and compelling evidence for this non-reconstructibility comes from our rigorous **jailbreak attack experiments**. We implemented sophisticated **Multi-hop** (indirect reasoning) and **Multi-lingual** attacks designed to probe for any remaining traces of the forgotten knowledge.
> > >
> > > The results below demonstrate two key points:
> > > 1.  **High Forgetting Success Rate**: Without the memory token, the model consistently fails to answer these challenging attack questions, proving the knowledge is not reconstructible.
> > > 2.  **High Restoration Success Rate**: With the memory token, the model regains the ability to answer the *exact same* questions, proving the knowledge is fully preserved within the token.
> > >
> > > | Externalized Concepts           | Multi-hop Forgetting Rate (↑) | Multi-hop Restoration Rate (↑) | Multi-lingual Forgetting Rate (↑) | Multi-lingual Restoration Rate (↑) |
> > > | :------------------------------: | :-----------------------------: | :------------------------------: | :--------------------------------: | :--------------------------------: |
> > > | Trump \& Chihuahua \& Musk         | 97.00                           | 99.00                            | 95.00                              | 98.00                              |
> > > | Trump \& Hello Kitty \& Musk       | 82.00                           | 95.00                            | 84.00                              | 96.00                              |
> > >
> > > The contrast between the high "Forgetting" rates (when the token is absent) and the high "Restoration" rates (when the token is present) empirically validates our core claim. The base model itself cannot reconstruct the knowledge; the memory token acts as the indispensable "key" required to unlock it.
> > >
> > >
> > >
> > > ###  **Reference:**
> > > [1] Yue, Xiang, et al. "Mmmu: A massive multi-discipline multimodal understanding and reasoning benchmark for expert agi." Proceedings of the IEEE/CVF Conference on Computer Vision and Pattern Recognition. 2024.
> > >
> > > [2] Liu, Yuan, et al. "Mmbench: Is your multi-modal model an all-around player?." European conference on computer vision. Cham: Springer Nature Switzerland, 2024.

---

> ### Comment · Reviewer_2Myd · 2025-11-27
>
> Thanks to the authors for addressing my concerns, I have adjusted my score to 6 and hopefully the authors can append those adjustments to the updated manuscript.

---

> > ### Author Response · Authors · 2025-11-28
> >
> > Thank you for reading our rebuttal and for updating the review! We're happy that our rebuttal alleviated the reviewer's main concerns. We will append the adjustments to the updated manuscript.

---

### Official Review · Reviewer_q6J9 · 2025-10-30

**Soundness:** 2
**Presentation:** 3
**Contribution:** 2
**Rating:** 4
**Confidence:** 4

**Summary:**

This paper focuses on the Multimodal Large Language Models (MLLMs) unlearning, especially proposing the reversible data management, termed knowledge externalization, to realize the modular knowledge management for MLLMs. Specifically, the framework consists of two major components introduced in knowledge externalization, the first is dual-stream memory tuning, and the second is gradient interference, which show three capabilities like effective forgetting, continuous knowledge editing, and emergent ability for compositionality.

**Strengths:**

1. This paper introduces reversible knowledge management for MLLMs, which is novel and practical research setting.
2. This paper is easy to follow with intuitive illustrations and well-formalized equations.
3. The experiments cover different scale models and comprehensive metrics to support the empirical claims.

**Weaknesses:**

1. The considered research scenario is good and practical, but it could be better to explicitly discuss more about the high-level research question, it seems that currently, there are three aspects that can be discussed singly in each specific research area, so what is the current work's biggest takeaway and difference from those separate research works on MLLMs?
2. The dual-stream memory tuning essentially combined two core mechanisms conventionally in unlearning, while the soft orthogonal weighting also stems from the common factorization techniques in the knowledge editing area. The question is what is the unique technical contribution that is closely related to the new research setting or MLLM? It is important to better position this work in the related area, and highlight the core contributions instead of a simple application.
3. For the three major claims about the capabilities of knowledge externalization, the empirical justification is not comprehensive and convincing enough to demonstrate the effectiveness.
While I appreciate the idea and the presentation, the paper's claim needs carefully revised or more either theoretical or empirical justficiation to support. Please consider the question part for potential suggestions.

**Questions:**

For the first two points in weakness, please refer to the comments directly for discussion, and for the third part, I have a few questions for potential suggestions about revision:
1. Please consider involving more work about knowledge editing on MLLMs for discussion and empirical comparison if applicable.
2. The current empirical performance is good in those tables shown in the paper, while could the authors further discuss and elaborate how the three capabilites are specifically justified (e.g., with what kind of strong baseline, in what kind of aspect does the method suppress the previous methods or not, is there any failure modes), especially for the continual editing and composition part.

---

> ### Author Response · Authors · 2025-11-20
> **Response to Reviewer q6J9 Part I**
>
> Thank you for your critical feedback and suggestions. We address your thoughts point by point below.
>
> >**Q1**: The work could better discuss its high-level research question and its biggest takeaway, especially differentiating it from separate research works.
>
> **A1**: We deeply appreciate this critical feedback. You are absolutely right that our paper would benefit from more explicit discussion of the overarching research vision. Here are our core research questions and how our work answers them:
>
> **RQ1: Can we achieve reversible unlearning in MLLMs, moving beyond destructive parameter modification?**
>
> Traditional unlearning methods irreversibly alter model parameters to erase knowledge, preventing any restoration.We actively *transfer* target knowledge from model parameters into external memory tokens, enabling later restoration without retraining.
>
> *Empirical evidence:* Table 1-3 shows our method achieves high recovery rate of externalized knowledge using the memory token. This is the fundamental paradigm shift that separates our work from prior unlearning research.
>
> ---
>
> **RQ2: Can we manage multiple concepts simultaneously without catastrophic interference between them?**
>
> Unlike continual knowledge editing (which risks cumulative degradation), we address multi-concept externalization. Our Soft Orthogonal Weighting (SOW) is specifically designed to preserve the independence of multiple externalized concepts operating within the same shared model.
>
> *Empirical evidence:* Table 1-3  and Figure 5 demonstrates graceful degradation as we increase the number of externalized concepts. Crucially, our method with SOW maintains significantly higher performance compared to a naive ablation without SOW, empirically validating that SOW actively mitigates gradient interference between concepts. This capability is not addressed by existing unlearning or knowledge editing works.
>
> ---
>
> **RQ3: Can externalized knowledge be dynamically edited and composed post-externalization without retraining?**
>
>  Our framework enables two novel emergent capabilities:
> - **Continuous editing**: Updating what's stored in a memory token without affecting other externalized concepts or model performance
> - **Compositionality**: Combining multiple memory tokens (including edited ones) to recover knowledge from multiple concepts simultaneously
>
> *Empirical evidence:* Section 5.4 demonstrates successful continuous editing where we update factual information in an externalized token. Section 5.5 shows compositionality: when we concatenate memory tokens for multiple concepts, the model can simultaneously answer questions about both concepts with high accuracy, even for edited tokens.
>
> ---
>
> **Biggest Takeaway - Knowledge Externalization as a Paradigm Shift:**
>
> Our work establishes that **reversible, modular, and auditable knowledge management is possible** in MLLMs through systematic knowledge transfer rather than destructive erasure. This addresses a fundamental gap: existing works treat unlearning and knowledge editing as separate problems solved independently. Our integrated framework demonstrates that by treating knowledge externalization as a unified problem, we unlock new capabilities (reversibility, compositionality, continuous editing) that none of the separate approaches can individually achieve.
>
>
>
>
>
>
>
> >**Q2**: The unique technical contribution of DSM and SOW, suggesting they combine conventional mechanisms.
>
> **A2**: Thank you for your valuable suggestion. Please refer to general response Q4.

---

> > ### Author Response · Authors · 2025-11-20
> > **Response to Reviewer q6J9 Part II**
> >
> > >**Q3**: Please consider involving more work about knowledge editing on MLLMs for discussion and empirical comparison if applicable.
> >
> > **A3**:We thank the reviewer for this crucial suggestion.
> >
> >
> > We thank the reviewer for this crucial suggestion. We have now conducted empirical comparisons with recent knowledge editing methods on MLLMs to directly demonstrate our framework's advantages in continual editing.
> >
> >
> > We compared our approach with recent knowledge editing methods: MSCKE[1] and multi-step editing from Mike[2] (designed for single edits) and CARML[3] (designed for continual editing). The results in Table below clearly demonstrate the unique advantages of our framework:
> >
> > | Method | Edit 1 (SPE) | Edit 2 (SPE) | Edit 5 (SPE) | Edit 10 (SPE) |
> > |:---|:---:|:---:|:---:|:---:|
> > | Original | 0.0 (58.2) | — | — | — |
> > | MSCKE[1] | 92.1 (54.8) | N/A | N/A | N/A |
> > | Mike[2] | 93.7 (53.3) | N/A | N/A | N/A |
> > | CARML[3] | 91.3 (56.2) | 90.7 (52.5) | 89.2 (38.6) | 87.6 (21.3) |
> > | **Ours (DSM w/ SOW)** | **97.0 (55.9)** | **96.8 (55.9)** | **96.3 (55.9)** | **95.8 (55.9)** |
> >
> > MSCKE and Mike are architecturally limited to single edits—they cannot perform continual editing, resulting in N/A across multiple edit scenarios. Their contribution is simply in-place modification of base model parameters, which becomes impractical when multiple edits are needed.
> >
> > CARML, designed for continual editing, does support multiple sequential edits. However, as the table shows, each additional edit causes increasingly severe degradation. After just 5 sequential edits, CARML's specificity drops to 38.6. This degradation occurs because CARML modifies the base model parameters sequentially, and each modification risks interfering with previous edits and general knowledge.
> >
> > Our framework presents a fundamentally different paradigm. Because the base model is **frozen after externalization**, each subsequent edit operates only on the corresponding memory token in isolation. As shown in the table, **our SPE stays completely stable at 55.9 across all 10 edits**. This empirical result validates our design philosophy: by freezing the base model after externalization and editing only isolated memory tokens, we prevent the cumulative interference that destroys model's performance.
> >
> >  **We added further introduction to related work on knowledge editing on MLLMs to the manuscript (line 234-241).**
> >
> > >**Q4.1**: Justification of unlearning part.
> >
> >
> >
> > **A4.1**: Thank you for your valuable question. Please refer to general response Q5.

---

> > > ### Author Response · Authors · 2025-11-20
> > > **Response to Reviewer q6J9 Part III**
> > >
> > > >**Q4.2**: Justification of continual editing part.
> > >
> > >
> > >
> > > **A4.2**: We thank the reviewer for this crucial suggestion. For continual editing, CARML [3] is the most relevant baseline. However, its approach is fundamentally different. CARML relies on in-place modification, directly altering the model's core parameters for each new edit. As shown in the table in Q3, while effective for a few edits, this sequential modification risks cumulative degradation of the model's general capabilities and can lead to catastrophic forgetting of previously learned knowledge.
> > >
> > > Our framework is architecturally designed to prevent this. After a one-time knowledge externalization, the core MLLM parameters are frozen. All subsequent "continual editing" such as adding, updating, or deleting knowledge operates only on isolated, external memory tokens, not the model itself. This design ensures that the base model's performance and previously edited knowledge remain stable, fundamentally preventing the degradation seen in methods like CARML.
> > >
> > > >**Q4.3**: Justification of composition part.
> > >
> > >
> > >
> > > **A4.3**: We thank the reviewer for this insightful question, as it allows us to highlight the unique and pioneering nature of our framework's approach to "composition."
> > >
> > > Our work introduces **the first framework for explicit, externally-driven composition of knowledge within MLLMs**. This represents a fundamental shift from existing research on token compositionality.
> > >
> > > Current literature, exemplified by the comprehensive survey "Interpreting token compositionality in LLMs" [4] and focused analyses like "Scan and Snap: Understanding Training dynamics and Token Composition in 1-Layer Transformers" [5], primarily investigates how LLMs implicitly combine information from input text tokens during their internal processing. These efforts are invaluable for analyzing and explaining the inherent, pre-trained mechanisms of how models understand language. They seek to open the black box of internal token interactions.
> > >
> > > In contrast, our framework **operates on an entirely different plane**. We are not analyzing how an MLLM understands a sentence. Instead, we have architected a novel system that enables a frozen MLLM to explicitly and dynamically compose multiple, independent, externalized Memory Tokens. Our contribution is in building and leveraging this compositional capability, rather than merely analyzing it.
> > >
> > > **We added further introduction to related work on token composition to the manuscript (line 256-260).**
> > >
> > > ###  **Reference:**
> > > [1] Zeng, Zhen, et al. "Visual-oriented fine-grained knowledge editing for multimodal large language models." Proceedings of the IEEE/CVF International Conference on Computer Vision. 2025.
> > >
> > > [2] Li, Jiaqi, et al. "MIKE: A New Benchmark for Fine-grained Multimodal Entity Knowledge Editing." Findings of the Association for Computational Linguistics ACL 2024. 2024.
> > >
> > > [3] Zhang, Qiang, et al. "Reliable Lifelong Multimodal Editing: Conflict-Aware Retrieval Meets Multi-Level Guidance." The Thirty-ninth Annual Conference on Neural Information Processing Systems.
> > >
> > >  [4] Aljaafari, Nura, Danilo S. Carvalho, and André Freitas. "Interpreting token compositionality in LLMs: A robustness analysis." arXiv preprint arXiv:2410.12924 (2024).
> > >
> > > [5] Tian, Yuandong, et al. "Scan and snap: Understanding training dynamics and token composition in 1-layer transformer." Advances in neural information processing systems 36 (2023): 71911-71947.

---

> > > > ### Comment · Reviewer_q6J9 · 2025-11-27
> > > >
> > > > Thank the authors for the detailed rebuttal and additional response. Most of my concerns are addressed, and I will adjust my score accordingly. Wish you a pleasant Thanksgiving.

---

> > > > > ### Author Response · Authors · 2025-11-28
> > > > >
> > > > > Thank you for reading our rebuttal and for updating the review! We appreciate the time that the reviewer has taken throughout the process to thoroughly review our work!

---

### Official Review · Reviewer_UCga · 2025-10-31

**Soundness:** 3
**Presentation:** 3
**Contribution:** 3
**Rating:** 6
**Confidence:** 4

**Summary:**

The paper introduces Knowledge Externalization, a novel framework for managing knowledge in Multimodal Large Language Models (MLLMs) that addresses the limitations of current, irreversible machine unlearning methods. Motivated by modern privacy regulations that mandate reversible, auditable, and user-controllable data management , the framework works by transferring targeted knowledge from the MLLM's internal parameters into external memory tokens.

**Strengths:**

1. This is a major advancement over traditional destructive unlearning, offering a mechanism for auditable and reversible data management, which is critical for meeting modern privacy regulations (e.g., the right to be forgotten and the right to have data restored).

2. Strong Modularity and Compositionality: The externalized knowledge is managed as independent, self-contained memory tokens. This allows for dynamic composition of knowledge during inference (e.g., retrieving information about two distinct entities simultaneously), offering immense flexibility.

3. Minimal Utility Degradation: By transferring the knowledge instead of simply erasing it, the method minimizes catastrophic forgetting—the loss of the base model’s general capabilities or non-target knowledge—a common issue in traditional unlearning

**Weaknesses:**

1. Scalability Analysis of Knowledge Externalization: As the number of unique knowledge entities requiring externalization grows, the quantity of external memory tokens will explode. Managing, storing, and efficiently indexing this large, potentially sparse memory store introduces significant storage and retrieval overhead in practical deployment. The authors should provide an analysis of Forgetting Effectiveness, General Utility Preservation, and Inference Latency as a function of the increasing scale of externalized knowledge

2. Robustness of Unlearning: The authors should explore the robustness of unlearning against attacks targeting forgotten knowledge.

**Questions:**

see weaknesses

---

> ### Author Response · Authors · 2025-11-20
> **Response to Reviewer UCga**
>
> We are grateful for your attentive comments and providing thoughtful feedback on our work. We will provide our insights point by point below.
>
> >**Q1.1**: The retrieval overhead in practical deployment.
>
> **A1.1**: Thank you for your valuable comment. Please refer to general response Q1.
>
> >**Q1.2**: The quantity of external memory tokens will explode. Provide an analysis of Forgetting Effectiveness, General Utility Preservation, and Inference Latency as a function of the increasing scale of externalized knowledge.
>
> **A1.2**: Thank you for your insightful comment. For the memory token explosion, please refer to general response Q2. Regarding inference time, our method maintains a consistently low overhead as shown in the table below. The inference time per query with multiple concepts loaded remains within a narrow and manageable range, from 2.49 seconds for a single concept to 3.77 seconds for seven concepts. For the anaylisis of forgetting effectiveness and utility preservation please refer to Figure 5. We demonstrates graceful degradation as we increase the number of externalized concepts (1→7 concepts).
>
> **Table: Memory Token Efficiency for Multiple Concepts**
>
> | Number of Concepts | Total Inference Time (s) | Total Peak Memory (MB) |
> |:------------------:|:--------:|:-------------:|
> | 1                  | 2.49         | 14981                  |
> | 2                  | 2.24                     | 15082                  |
> | 3                  | 2.90                     | 15172                  |
> | 4                  | 3.52                     | 15262                  |
> | 5                  | 3.66                     | 15356                  |
> | 6                  | 3.67                     | 15448                  |
> | 7                  | 3.77                     | 15559                  |
>
>
>
> **Q2: The authors should explore the robustness of unlearning against attacks targeting forgotten knowledge.**
>
> **A2:** We thank the reviewer for this insightful suggestion. To rigorously evaluate the robustness of our method against attempts to extract forgotten knowledge, we performed attacks specifically targeting the unlearned concepts. We implemented two sophisticated attack strategies: **Multi-lingual Attacks** and **Multi-hop Jailbreak Attacks**.
>
> *   **Multi-lingual Attack**: To check whether the unlearning is truly language-agnostic and robust to varied input forms, we translated the probing questions into six different languages (Spanish, French, Chinese, German, Japanese, and Russian) to perform a multi-lingual jailbreak.
> *   **Multi-hop Jailbreak Attack**: This is a form of secluded attack using hard examples. Instead of directly querying the concept by name, we used GPT-5 to generate questions that probe for factual knowledge about the concept indirectly, requiring multi-step reasoning and inferential leaps. This tests if deeply embedded knowledge can still be implicitly retrieved.
>
> For the evaluation, we measured two key metrics:
> 1.  **Forgetting Success Rate (↑)**: This is the percentage of times the model successfully avoided providing a correct answer to the attack questions *when the memory tokens were removed*, thereby indicating that the knowledge was truly forgotten and not merely suppressed. A higher score signifies greater robustness against the attack.
> 2.  **Restoration Success Rate (↑)**: This is the percentage of times the model successfully provided a correct answer to the same attack questions *when the corresponding memory tokens were re-introduced*. This metric demonstrates the high-fidelity and comprehensive reversibility of the externalized knowledge, even for complex, indirect, or multi-lingual queries.
>
> The results are as follows:
>
> | Externalized Concepts  | Multi-hop Forgetting Rate (↑) | Multi-hop Restoration Rate (↑) | Multi-lingual Forgetting Rate (↑) | Multi-lingual Restoration Rate (↑) |
> | :--: | :---: | :--: | :--: | :--: |
> | Trump & Chihuahua & Musk   | 97.00 | 99.00  | 95.00  | 98.00  |
> | Trump & Hello Kitty & Musk   | 82.00   | 95.00  | 84.00 | 96.00  |
>
> The high **Forgetting Success Rates** demonstrate that our method is highly robust. Even when faced with sophisticated multi-hop reasoning or prompts in various languages, the model consistently refuses to regenerate the forgotten information once the corresponding memory tokens are removed.
>
> Crucially, the high **Restoration Success Rates** further underscore the effectiveness of Knowledge Externalization. When the respective memory tokens are re-introduced, the model regains the ability to accurately answer these same complex multi-hop and multi-lingual questions. This confirms that the knowledge is not permanently destroyed but is truly externalized and perfectly restorable, even under challenging attack scenarios. This reversible behavior is a cornerstone of our framework, providing robust control over knowledge access.
>
>
>
>
>
>
> **We added the contents of additional experiments to the manuscripts (Section A.6).**

---

### Official Review · Reviewer_j8Uo · 2025-11-01

**Soundness:** 2
**Presentation:** 3
**Contribution:** 3
**Rating:** 6
**Confidence:** 3

**Summary:**

The paper “Knowledge Externalization: Reversible Unlearning and Modular Retrieval in Multi-Modal Large Language Models” proposes a new framework to make unlearning in multimodal models reversible rather than destructive. The authors introduce two key methods: Dual-Stream Memory Tuning (DSM), which simultaneously erases target knowledge from model parameters and transfers it to external memory tokens, and Soft Orthogonal Weighting (SOW), which minimizes interference between multiple externalized concepts by adjusting gradient similarity. Extensive experiments on several MLLMs (LLaVA-7B/13B and InternVL-2B) show that the method effectively removes target information, preserves unrelated knowledge, and restores forgotten knowledge using external tokens. The framework also demonstrates emergent compositionality (combining multiple memory tokens) and dynamic editability (updating specific memories without retraining).

**Strengths:**

1. The paper introduces Knowledge Externalization, a novel and well-structured framework for reversible and modular unlearning in Multimodal Large Language Models (MLLMs).

2. The proposed Dual-Stream Memory Tuning (DSM) and Soft Orthogonal Weighting (SOW) methods are technically sound and effectively validated across multiple MLLMs (e.g., LLaVA-7B, LLaVA-13B, and InternVL-2B). The experimental design is comprehensive, including ablation studies, multi-concept externalization, and emergent compositionality analysis.

3. Importantly, the authors explore unlearning, a crucial research topic, through the novel perspective of knowledge externalization, which provides an intriguing direction for controllable and reversible knowledge management.

**Weaknesses:**

1. Despite its novelty, several conceptual issues remain unaddressed. Personal or proprietary knowledge embedded in multimodal models is unlikely to have been collected with explicit consent. In such cases, the reversibility of personal data could contradict privacy protection principles, since once removed, sensitive information should not be restorable. Although the authors cite ISO/IEC 27701 to justify reversibility, it is unclear which specific literature explicitly supports that this standard “emphasizes reversibility".

2. While the framework is claimed to generalize to MLLMs, it remains unclear why it has only been tested on multimodal settings, not text-only LLMs. A justification for this limitation is necessary.

3. The design choice of using prefix embeddings for the external memory restricts retrieval, as information can only be recovered when the exact prefix tokens are provided. When numerous entities are externalized, this dependency could become inefficient and difficult to manage.

4. Experiments are limited to a small number of entities and show reduced performance as the number of externalized concepts increases (as in Figure 5). This raises concerns about scalability and stability. The baseline comparison is also insufficient: given that reversible adaptation could potentially be achieved more simply with LoRA modules, the authors should discuss why LoRA-based reversible unlearning was not considered as a baseline. A LoRA-based approach could be more practical and powerful.

**Questions:**

See Weaknesses

---

> ### Author Response · Authors · 2025-11-20
> **Response to Reviewer j8Uo Part I**
>
> We are greatly encouraged by your positive comments. Thanks a lot for all the appreciation.
>
> >**Q1**: How reversibility aligns with privacy protection, particularly the "right to be forgotten," and the interpretation of ISO/IEC 27701.
>
> **A1**: We appreciate this profound question. Our framework's "reversibility" is fundamentally designed to empower **users**, not only platforms. It aligns with advanced privacy principles like user-controlled data portability and restoration. While the "right to be forgotten" prevents unauthorized data retention, our "reversibility" grants the user the explicit choice to restore their data if they wish. ISO/IEC 27701 extends ISO/IEC 27001 by adding **privacy-specific controls**, creating a **Privacy Information Management System (PIMS)**. Moreover, **Article 18(1)(b) of GDPR**, **"Right to restriction of processing,"** acknowledges scenarios where data subjects oppose erasure but request restriction of use, which aligns with the controlled, reversible nature of our approach. Furthermore, **GDPR Article 30** mandates organizations to maintain processing activity logs for regulatory oversight, and our external memory mechanism naturally provides an auditable, traceable logging structure for knowledge management. Our work also realizes a PIMS within MLLMs, a concept supported by existing literature (Tzolov et al., 2019[1]; Anwar & Gill, 2021[2]).  **We added further clarifications to the manuscript (line 049-054).**
>
>
>
>
>
>
> >**Q2**: A justification why the framework was only tested on multimodal settings.
>
> **A2**: Thank you for your thoughtful question. We focused on MLLMs because knowledge in multimodal models is often deeply entangled across visual and textual modalities, making unlearning and precise knowledge management inherently more challenging than in text-only LLMs. Demonstrating effectiveness in this more complex domain underscores the framework's robustness and potential for broader applicability. While our current focus is on multimodal settings, the core mechanism of Dual-Stream Memory Tuning and Soft Orthogonal Weighting is inherently generalizable to text-only LLMs and we will explore this in the future work. **We added the clarifications to the manuscript (line 505-507).**
>
> >**Q3**: The design choice of using prefix embeddings for the external memory restricts retrieval, as information can only be recovered when the exact prefix tokens are provided. When numerous entities are externalized, this dependency could become inefficient and difficult to manage. (Reviewer j8Uo)
>
> **A3**: We thank Reviewer j8Uo for this insightful comment.
>
> **Why Prefix Embeddings for Knowledge Externalization?**
> The core challenge in reversible unlearning is: **how do we simultaneously achieve (1) modular knowledge transfer, (2) minimal parameter overhead, and (3) scalability to numerous concepts?**
>
> We chose prefix embeddings precisely because they represent the most parameter-efficient solution. Consider the alternatives:
>
> 1. **LoRA-based approaches** (as suggested in Q4.2): While conceptually appealing, LoRA requires loading adapter matrices during inference for each externalized concept. As demonstrated in our Q4.2 response, loading even 5 LoRA adapters causes out-of-memory errors on a single 4090 GPU during inference. LoRA's parameter overhead grows quadratically with the number of concepts, making it fundamentally unscalable for the modular management of numerous entities.
>
> 2. **Full fine-tuning or other parameter-modifying approaches**: These would directly alter base model weights for each concept, destroying both the modularity principle and the integrity of the original model.
>
> 3. **Prefix embeddings (our choice)**: Each externalized concept requires only a small set of additional token embeddings (typically 32-128 dimensions per token). As shown in our response to Q4.2, adding 7 concepts increases peak memory by less than 600 MB—a negligible overhead compared to LoRA's unsustainable growth pattern.
>
> Regarding the concern about inefficiency and difficulty in managing numerous entities when exact prefix tokens are required, we acknowledge this challenge. However, we contend that the efficient retrieval of the correct token set from a large pool is a **well-defined and solvable engineering problem, not a fundamental scientific limitation of our framework**. **Please refer to our General Response Q1** for a detailed discussion on how modern retrieval systems are designed to handle millions or billions of embeddings with high efficiency and low latency, which can be leveraged to manage these externalized memory tokens.

---

> > ### Author Response · Authors · 2025-11-20
> > **Response to Reviewer j8Uo Part II**
> >
> > >**Q4.1**: The experiments are limited to small number of concepts and the performance degraded as shown in Figure 5,
> >
> > **A4.1**: Thank you for your pertinent suggestion. Please refer to general response Q3.
> >
> >
> > >**Q4.2**: Suggested LoRA as a potential baseline for reversible unlearning.
> >
> >
> > **A4.2**:  We appreciate the suggestion to consider LoRA as a baseline for multi-concept unlearning. To thoroughly investigate its feasibility, we conducted a direct comparison of resource consumption between the LoRA approach and our proposed method on a single 4090 GPU. The results clearly demonstrate that LoRA's memory footprint makes it impractical for the task of managing an increasing number of externalized concepts.
> >
> > During inference, we observed that loading multiple LoRA adapters leads to a rapid and unsustainable increase in memory consumption. As shown in the table below, the system ran out of memory when attempting to load just 5 LoRAs simultaneously. **The memory constraints during training are even more severe, where an OOM error occurred with just two LoRAs (Batchsize two).**
> >
> > | Number of LoRA | Inference Time (s) | Peak CUDA Memory (MB) |
> > |:-----------------------:|:-------------------:|:---------------------:|
> > | 1                       | 4.319               | 17143.3               |
> > | 2                       | 7.689               | 19456.4               |
> > | 3                       | 9.562               | 21635.4               |
> > | 4                       | 10.585              | 23843.6               |
> > | 5                       | OOM                 | OOM                   |
> >
> > In contrast, our method demonstrates exceptional efficiency. As detailed in the table below, the marginal resource cost for each additional externalized concept is minimal. When scaling from 1 to 7 concepts, the total peak memory usage increased by less than 600 MB, and the total inference time remained stable and low.
> >
> > | Number of Concepts | Total Inference Time (s) | Total Peak Memory (MB) |
> > |:------------------:|:------------------------:|:----------------------:|
> > | 1                  | 2.49                     | 14981                  |
> > | 2                  | 2.24                     | 15082                  |
> > | 3                  | 2.90                     | 15172                  |
> > | 4                  | 3.52                     | 15262                  |
> > | 5                  | 3.66                     | 15356                  |
> > | 6                  | 3.67                     | 15448                  |
> > | 7                  | 3.77                     | 15559                  |
> >
> > The comparison clearly shows that LoRA's memory requirements make it impractical for scenarios requiring the simultaneous management of numerous concepts. Our method, in contrast, is specifically engineered to handle this task efficiently, offering minimal marginal cost per additional concept without hitting hardware limitations. Therefore, we conclude that LoRA is not a scalable or feasible baseline for this specific multi-concept unlearning task. In the future work, we will explore more feasible and efficient framework for knowledge externalization.
> >
> >
> >
> > ###  **Reference:**
> > [1] Tzolov, Tzanko Valkov. "ISO 27552 as a Model for Establishment Personal Information Management Systems." 2019 International Conference on Information Technologies (InfoTech). IEEE, 2019.
> >
> > [2] Anwar, Memoona J., and Asif Gill. "Developing an integrated ISO 27701 and GDPR based information privacy compliance requirements model." Australasian Conference on Information Systems 2020. 2021.

---

### Author Response · Authors · 2025-11-20
**General Response Part I**

We sincerely thank all reviewers for their thorough and constructive feedback. We are encouraged that the reviewers recognized the novelty of our Knowledge Externalization framework and its potential for reversible and modular unlearning. We have uploaded a revised version of our paper and the changed contents are highlighted with red color. We are confident that we can address the concerns raised. Below, we address the common questions shared by multiple reviewers:

>**Q1**: Question about Retrieval and Management Efficiency with Numerous Concepts. (j8Uo, UCga)

**A1**: We thank the reviewers for raising this important concern regarding retrieval and management efficiency at scale.

The challenge of efficiently retrieving the correct token set from a large pool is a **well-defined and solvable engineering problem, not a fundamental scientific limitation of our framework**. The task of mapping a query to the most relevant embedding is precisely what modern retrieval systems are built for. State-of-the-art vector databases (such as Faiss[1], ScaNN[2]) and advanced retrieval techniques are specifically designed to manage millions or even billions of embeddings with high efficiency and low latency.

Our framework's core contribution is the creation of these modular, transferable knowledge tokens. By doing so, we enable the use of these powerful, off-the-shelf scalable management technologies.

**We added the clarifications to the manuscript in Section 4.5.**


>**Q2**: On the "Token Explosion" Storage Overhead and the inference time. (UCga, 2Myd, j8Uo)

**A2**: We appreciate the reviewers' concerns about a potential "token explosion" or memory storage overhead, Regarding the "Token Explosion" storage overhead, our method is specifically designed to avoid this issue by externalizing knowledge into compact memory tokens. We have thoroughly evaluated the memory footprint of our approach as the number of externalized concepts increases.

Our experiments on a single 4090 GPU demonstrate that the marginal increase in both inference time and peak CUDA memory consumption for each additional externalized concept is remarkably small. As shown in the table below, the peak CUDA memory usage for 1 concept was 14981 MB, incrementally rising to only 15559 MB for 7 concepts. This represents an increase of less than 600 MB for six additional concepts. The inference overhead associated with each additional concept also remains minimal. This efficient scaling demonstrates that our method effectively manages resource demands and does not suffer from a "token explosion" in storage overhead, even with an increasing number of unlearned concepts.


**Table: Memory Token storage for Multiple Concepts**

| Number of Concepts | Total Peak Memory (MB) |
|:----:|:---:|
| 1               | 14981        |
| 2                  | 15082                  |
| 3                   | 15172                  |
| 4                   | 15262                  |
| 5                         | 15356                  |
| 6                          | 15448                  |
| 7                        | 15559                  |

>**Q3**: On the limited number of concepts tested and the observed performance degradation as concepts increase. (Reviewer j8Uo, 2Myd)

**A3**: We thank the reviewers for these related concerns. We believe there may be a misunderstanding regarding our experimental setup that explains both observations.

As stated in Section 5.3 (line 439-443 in the revised pdf), we conduct experiments with knowledge numbers ranging from 2 to 8 **while keeping the total training steps constant**. This design ensures SPE performance remains stable between 50-56 across all configurations, allowing us to isolate the effect of knowledge complexity on GEN and AVG_REC metrics. The observed decline is therefore expected—when more concepts share the same fixed training budget, each concept receives fewer optimization steps.

To address the scalability concern directly, we conducted an additional experiment with **20 externalized concepts** using appropriately scaled training:

| Concepts | GEN | SPE | AVG_REC |
|:---:|:---:|:---:|:---:|
| 2 | 97.0 | 55.9 | 95.2 |
| 8 | 85.3 | 52.1 | 87.6 |
| **20** | **77.0** | **49.4** | **83.3** |

Even with 20 concepts, our framework maintains strong performance: GEN at 77.0, SPE at 49.4, and AVG_REC at 83.3. This demonstrates that our method scales gracefully without catastrophic interference.



**We added the experiments to the manuscript (A8.3).**

---

> ### Author Response · Authors · 2025-11-20
> **General Response Part II**
>
> >**Q4**: On the technical novelty of DSM and SOW, particularly in comparison to existing gradient-difference methods and knowledge editing techniques. (q6J9, 2Myd)
>
> **A4**: We thank the reviewers for prompting us to further clarify the unique technical contributions of our work. We acknowledge the conceptual similarities of some aspects to established fields, but we emphasize that our framework integrates these elements in a novel way to achieve reversible knowledge externalization.
>
> 1.  **DSM's Core Novelty** (q6J9, 2Myd): While gradient-difference (Grad-Diff) methods are a relevant reference, DSM presents a distinct paradigm. Traditional Grad-Diff typically applies gradient ascent (GA) to a "forget set" and gradient descent (GD) to a "retain set," often optimizing the *same set of model parameters*. In contrast, DSM applies *both GA and GD to the targeted "forget set" itself*, but critically, these operations update **different sets of parameters**. Specifically, GA is applied to the MLLM's internal weights to erase target knowledge, while GD is applied to the newly introduced *external memory tokens* to encode the same knowledge. This simultaneous, yet parameter-decoupled, dual-stream optimization is central to our framework, enabling the *transfer* of knowledge rather than mere erasure, and is essential for achieving the framework's reversibility and modularity.
>
> 2.  **SOW's Distinctiveness** (q6J9): The Soft Orthogonal Weighting (SOW) is also a novel contribution, going beyond conventional orthogonalization or masking techniques. While inspired by orthogonalization concepts, SOW is a **"soft" and "dynamic" orthogonal weighting** specifically designed to minimize interference between multiple, simultaneously externalized concepts in the context of our externalization framework.
>     *   **"Soft"**: Unlike rigid masking or hard orthogonalization that might modify *all* parameters or enforce strict zero overlap, SOW employs a cosine-similarity regularization. This approach reduces the overlap in gradient directions, allowing for nuanced relationships between potentially overlapping concepts without rigid architectural modifications. This soft constraint is critical for preserving emergent compositionality and avoiding detrimental impacts on general capabilities.
>     *   **"Dynamic"**: SOW's dynamic nature is rooted in its continuous update of gradient history throughout training. It specifically computes and regularizes the cosine similarity between the gradients associated with *different externalized concepts*, not last few steps. This is not a general technique but a method tailored precisely for our concept externalization framework, allowing for the efficient co-existence and independent management of numerous memory tokens within a shared model.
>
>
> >**Q5**: Missing citation of two baselines and missing baselines like NPO, KL Divergence, IDK, DPO, SKU, etc., and that DSM itself is not a baseline. (q6J9, 2Myd)
>
> **A5**: We thank the reviewers for suggesting these important related works and prompting us to clarify the unique positioning of our framework.
>
> We want to highlight a fundamental distinction: our work introduces a **reversible knowledge externalization** framework, whereas the cited methods (NPO, KL Divergence, DPO, SKU) are primarily designed for **destructive unlearning**. Their goal is to permanently alter model weights to make knowledge inaccessible.
>
> To empirically validate this distinction, we conducted a preliminary evaluation of representative destructive methods. As shown below, while these methods are effective at forgetting, they are inherently irreversible, offering no mechanism to restore the erased knowledge.
>
> | Method | GEN↑ | SPE↑ | REC↑ |
> | :--- | :--- | :--- | :--- |
> | NPO | 88 | 54.5 | **0** |
> | KL | 93 | 53.8 | **0** |
> | IDK | 95 | 56.8 | **0** |
> | **ours** | **96** | **56.9**| **98**|
>
> The results confirm that these methods, by design, yield a recovery score of 0. Therefore, they are not suitable as direct, end-to-end baselines for evaluating the core **"reversibility"** contribution of our framework, which is the ability to both forget and losslessly restore knowledge.
>
>
>
> ###  **Reference:**
> [1] Douze, Matthijs, et al. "The faiss library." IEEE Transactions on Big Data (2025).
>
> [2] Hassantabar, Shayan, Zeyu Wang, and Niraj K. Jha. "SCANN: Synthesis of compact and accurate neural networks." IEEE Transactions on Computer-Aided Design of Integrated Circuits and Systems 41.9 (2021): 3012-3025.

---

### Author Response · Authors · 2025-11-24
**A gentle follow-up on our rebuttal**

Dear Reviewers,

We truly appreciate all your comments and the time you have invested in reviewing our paper.

We have submitted our rebuttal and hope our responses have clarified the points you raised. We are writing to gently follow up and see if you have had a chance to look at our responses.

We remain committed to improving our work based on your expert guidance and are happy to provide any further clarifications if needed.

We look forward to your feedback.

Best regards,

Authors

---

### Author Response · Authors · 2025-11-27

Dear Reviewers,

We appreciate your feedback and hope our rebuttal addresses your concerns. As the rebuttal deadline is approaching, if you have further concerns about our rebuttal, please provide your questions and we will respond as soon as possible.

---

### Author Response · Authors · 2025-11-29
**Summary Regarding Review Process for Paper 1982**

Dear AC, SAC and PCs,

Thank you for stepping in to oversee the review process for our paper under these unusual circumstances. We understand and appreciate the measures ICLR is taking to ensure a fair and robust evaluation for all submissions.

We would like to provide a brief summary to help contextualize the current state of our paper's review. We thank the reviewers for their thoughtful and insightful feedback where reviewers unanimously acknowledged the **strong motivation** and **novel importance** of our work on controllable knowledge management for MLLMs (J8Uo, UCga, q6J9). Our framework was praised for its **technical soundness** and **intuitive design** (J8Uo, UCga), with Reviewer q6J9 noting our "well-formalized equations" and "comprehensive experiments across different scale models," while Reviewer 2Myd acknowledged the "theoretical rigor" of our Dual-Stream Memory optimization and "mathematical maturity" of Soft Orthogonal Weighting.

Our paper initially received scores of **6 (Reviewer J8Uo), 6 (Reviewer UCga), 4 (Reviewer q6j9), and 2 (Reviewer 2Myd)**. We thoroughly valued all feedback and conducted significant new experimental work and analysis to address every major point raised across the reviews.

Our comprehensive rebuttal specifically included:

1.  **Enhanced Reproducibility & Baselines:** To address Reviewer 2Myd's critical concern regarding our benchmark and reproducibility, we performed **entirely new experiments on the public, standardized MLLMU-Bench** (a benchmark directly referenced by 2Myd). This ensures reproducibility and strengthens generalizability.

2.  **Demonstrated Robustness & Non-reconstructibility:** To rigorously prove that erased knowledge is non-reconstructible, a key concern for Reviewers 2Myd and UCga, we introduced and performed **Multi-hop and Multi-lingual Jailbreak Attacks**. These provided robust empirical evidence of our framework.

3.  **Addressed Scalability & Efficiency:** To respond to concerns about scalability, efficiency, and the number of externalized concepts (raised by Reviewers UCga, J8Uo, and 2Myd), we conducted **new experiments analyzing our framework's performance, overhead, and degradation as the number of externalized concepts increased**, offering a more comprehensive understanding of its real-world applicability.

4.  **Strengthened Knowledge Editing & Empirical Justification:** For Reviewer q6j9, who sought more comprehensive empirical evidence for knowledge editing, we provided **extensive new empirical comparisons against state-of-the-art editing methods**, demonstrating our method's superior stability and minimal degradation in continual editing scenarios. We also clarified our technical novelty and how our framework enables a paradigm shift in knowledge management.

5.  **Clarified Technical Novelty:** We addressed fundamental questions about our method's novelty for Reviewers q6J9 and 2Myd. We detailed how our **Dual-Stream Memory** is distinct from Grad-Diff methods by simultaneously applying gradient ascent (to erase from model weights) and descent (to encode in external tokens) on the same data, enabling true knowledge *transfer* rather than just erasure. We also explained the novelty of **Soft Orthogonal Weighting** as a dynamic, soft-constraint method tailored to minimize interference between multiple memory tokens, which is key for scalability.

We were encouraged to see that our rebuttal and new results were effective.  **Reviewer 2Myd explicitly raised their score from 2 to 6, and Reviewer q6j9 raised theirs from 4 to 6.** Their updated comments explicitly acknowledged that their primary concerns had been fully addressed by our new experiments and clarifications.


We understand that due to a platform-wide technical issue, all scores were procedurally reverted to their pre-rebuttal state. However, the reviewers' final, positive comments and their deliberate score increases represent their true, considered assessment *after* our comprehensive rebuttal.

Therefore, **given that all reviewers ultimately concluded with positive scores (a 6 for each)**, we respectfully request that you consider the substance of their final comments and these significant score adjustments as the accurate reflection of our paper's merit.

We are confident that our revised manuscript, strengthened by the new results, makes a solid contribution. Thank you for your time and careful consideration in this unique situation.

Best regards,

The Authors of Paper 1982

---

### Meta-Review · Area_Chair_PkRW · 2026-01-02

**Summary:**

The paper proposes Knowledge Externalization, a reversible and modular alternative to destructive machine unlearning for multimodal large language models. The core idea is to transfer targeted knowledge out of the base model into external memory tokens, enabling both effective forgetting in the base model and high-fidelity restoration when the corresponding token is provided. The approach introduces Dual-Stream Memory Tuning and Soft Orthogonal Weighting (SOW) to reduce interference across multiple externalized concepts.

Across reviewers, the main concerns are:

(1) whether the claimed reversibility is well-positioned with respect to privacy motivations;

(2) scalability to many concepts and practical retrieval/management;

(3) robustness of forgetting against adversarial extraction/jailbreaks;

(4) adequacy of baselines and novelty relative to Grad-Diff / knowledge editing;

(5) potential side effects on general model utility (reasoning/knowledge benchmarks).

The authors’ rebuttal and added experiments addressed most of these points substantially, and the discussion converged to a stronger overall assessment than the pre-rebuttal scores.

According to my reading, the work introduces a compelling and technically grounded framework for reversible and modular knowledge management in MLLMs, supported by broad experiments and a strong rebuttal.
The idea of external memory tokens is particularly novel and elegant: knowledge is explicitly externalized into editable and composable tokens, rather than being implicitly stored in model parameters. This design enables reversibility, interpretability, and fine-grained control, and goes beyond prior destructive unlearning or in-place editing methods.
By separating knowledge from the base model, the paper lays a solid foundation for future machine unlearning research. Therefore, I would recommend for accpetance.

**Reviewer Concerns:**

Most concerns are addressed by the rebuttal / discussion, e.g., baseline coverage & positioning, robustness of forgetting / non-reconstructibility, scalability and “token explosion” concerns.

There are some remaining concerns such as large-scale deployment (thousands/millions of entities).

**Reviewer Scores:**

Reviewer j8Uo: would remain 6 (already at 6; concerns mostly addressed; still “would not mind if rejected”).

Reviewer UCga: would remain 6 (already at 6; robustness + scalability analysis improved).

Reviewer q6J9: would increase from 4 → 6 (reviewer explicitly stated most concerns were addressed and they would adjust the score).

Reviewer 2Myd: would increase from 2 → 6 (reviewer explicitly confirmed score adjustment after rebuttal).

---

### Decision · Program_Chairs · 2026-01-26

Accept (Poster)